# Single-cell profiling reveals heterogeneity and functional patterning of GPCR expression in the vascular system

H. Kaur[1], J. Carvalho[1], M. Looso[2], P. Singh[1], R. Chennupati[1], J. Preussner[2], S. Günther[3], J. Albarrán-Juárez[1], D. Tischner[1], S. Classen[4], S. Offermanns[1,5] & N. Wettschureck[1,5]

G-protein-coupled receptor (GPCR) expression is extensively studied in bulk cDNA, but heterogeneity and functional patterning of GPCR expression in individual vascular cells is poorly understood. Here, we perform a microfluidic-based single-cell GPCR expression analysis in primary smooth muscle cells (SMC) and endothelial cells (EC). GPCR expression is highly heterogeneous in all cell types, which is confirmed in reporter mice, on the protein level and in human cells. Inflammatory activation in murine models of sepsis or atherosclerosis results in characteristic changes in the GPCR repertoire, and we identify functionally relevant subgroups of cells that are characterized by specific GPCR patterns. We further show that dedifferentiating SMC upregulate GPCRs such as *Gpr39, Gprc5b, Gprc5c* or *Gpr124*, and that selective targeting of *Gprc5b* modulates their differentiation state. Taken together, single-cell profiling identifies receptors expressed on pathologically relevant subpopulations and provides a basis for the development of new therapeutic strategies in vascular diseases.

[1] Department of Pharmacology, Max Planck Institute for Heart and Lung Research, Ludwigstr 43, 61231 Bad Nauheim, Germany. [2] ECCPS Bioinformatics Facility, Max Planck Institute for Heart and Lung Research, Ludwigstr 43, 61231 Bad Nauheim, Germany. [3] ECCPS Deep sequencing platform, Max Planck Institute for Heart and Lung Research, Ludwigstr 43, 61231 Bad Nauheim, Germany. [4] Harvey Vascular Centre, Kerckhoff-Klinik, Benekestraße 2–8, 61231 Bad Nauheim, Germany. [5] Medical Faculty, J.W. Goethe University, Theodor-Stern-Kai 7, 60590 Frankfurt, Germany. Correspondence and requests for materials should be addressed to N.W. (email: Nina.Wettschureck@mpi-bn.mpg.de).

G-protein-coupled receptors (GPCRs) are the largest family of transmembrane receptors in eukaryotes; they transduce signals of numerous physicochemical stimuli including neurotransmitters, hormones, local mediators, metabolic or olfactory cues, and light[1]. The human genome encodes ∼800 GPCRs, the majority of them being olfactory receptors. Among the 367 non-olfactory GPCRs are still ∼150 orphan receptors, that is, GPCRs for which ligand and function are still unknown[2–4]. GPCRs have regulatory functions in almost all organ systems, and dysregulation of GPCR signalling has been implicated in the pathogenesis of many diseases[5–9]. The unique combination of diversity and specificity within the GPCR family, together with the fact that they are readily targetable by exogenous agonists and antagonists, has made GPCRs a most successful group of drug targets[4]. In the vascular system, GPCRs modulate critical parameters such as vessel tone or endothelial permeability[10–12]; pharmacological modulation of receptors for angiotensin II, catecholamines or histamine are therefore crucial in the therapy of arterial hypertension or allergic fluid extravasation, respectively. However, although the market share of GPCR-targeting drugs is with 30–40% high, the overall number of targeted GPCR is with ∼30 rather low, suggesting that the potential of GPCRs as drug targets is not yet fully exploited[4].

The search for new GPCR-based therapies encompasses various strategies, among them the identification of new biased or allosteric ligands at GPCRs with known function and ligand or the deorphanization of GPCRs for which ligands are not yet known[13,14]. A third approach is the identification of new pathophysiologically relevant functions for specific GPCRs, in particular for orphan receptors. One useful strategy in the identification of new GPCR functions is the ever more detailed expression analysis in functionally relevant cell subpopulations, for example, in subgroups of vascular cell types. While GPCR expression has been studied in bulk cDNA of whole vessels or cultured vascular cells[15,16], our knowledge about GPCR expression in freshly isolated vascular cell types, in particular on the single-cell level, is insufficient. This is a major limitation, since current interpretations of expression data rely on the assumption that all cells of a given population are equal, or at least comparable, with respect to their GPCR repertoire. In contrast to this, studies in other fields indicate that gene expression is rather heterogeneous *in vivo*[17,18], and it is crucial for further development of GPCR-based therapeutic strategies to understand whether this is also true for GPCR expression, and if so, whether certain GPCRs are associated with functional subgroups within a given cell population[19]. Activated EC, for example, upregulate adhesion molecules and proinflammatory mediators and are crucial for the pathogenesis of inflammatory vascular disease[20]; whether this inflamed subpopulation of ECs is endowed with a specific GPCR repertoire, is unclear. Also vascular SMC undergo phenotypical changes in response to noxious stimuli, resulting in a dedifferentiated state characterized by downregulation of contractile proteins and upregulation of pro-proliferative, pro-migratory and proinflammatory markers as well as enhanced matrix synthesis[21,22]. Understanding in what way the GPCR repertoire differs between healthy and diseased vascular cells will be crucial to tailor new therapeutic strategies selectively targeting the latter.

To address this issue, we used here a microfluidic-based system for single-cell reverse transcription polymerase chain reaction (RT-PCR) to determine GPCR expression in individual freshly isolated EC and SMC. We found GPCR expression to be highly heterogeneous in all cell types and describe characteristic changes in single-cell GPCR repertoires in response to inflammatory activation. Furthermore, we identify functionally relevant subgroups of cells that are characterized by specific GPCR patterns and show examples of how this information can be used to modulate cell differentiation.

## Results

**Array design and quality control.** To narrow down the spectrum of GPCRs to be tested on the single-cell level, we determined GPCR expression in bulk RNA of vascular EC and SMC. Of 424 tested GPCRs, 122 GPCRs were expressed in at least one of these cell types (Supplementary Fig. 1A). For these 122 GPCRs, as well as for 32 additional GPCRs, a primer array for single-cell expression analysis was designed (Table 1). In addition, 13 genes

**Table 1 | Overview over the genes included in the array.**

| | Genes included | No. |
|---|---|---|
| GPCRs | – Adrenergic: *Adra1a, Adra1b, Adra1d, ~~Adra2c~~, ~~Adrb1~~, ~~Adrab2~~*<br>– Chemokine: *Ccbp2/Ackr2, Ccr10, Ccr1, Ccr2, Ccr3, Ccr4, Ccr5, Ccr6, Ccr7, Ccr8, Ccr9, Ccrl1, Ccrl2, Cx3cr1, Cxcr1, Cxcr2, Cxcr3, Cxcr4, Cxcr5, Cxcr6, Cxcr7, Xcr1*<br>– Lysophospholipid: *Lpar1, Lpar2, Lpar3, Lpar4, ~~Lpar6~~, S1pr1, ~~S1pr2~~, S1pr3, ~~S1pr4~~*<br>– Miscellaneous: *Adcyap1r1, Adora1, Adora2a, Agtr1a, ~~Aplnr~~, Avpr1a, ~~Bdkrb1~~, Bdkrb2, C3ar1, C5ar1, Calcrl, Chrm2, Chrm3, Cmklr1, Cnr2, Crhr2, Cysltr1, Ednra, Ednrb, Fpr1, Fpr2, Gabbr1, Glp1r, Glp2r, ~~Hrh1~~, Hrh2, ~~Htr1b~~, Htr2a, ~~Niacr1~~, Npy1r, Olfr78, Prokr1, Ptafr, Pth1r, Smo, ~~Sstr4~~, Sucnr1, ~~Tas2r126~~, ~~Tas2r135~~, ~~Tas2r143~~, Vipr1, Vipr2*<br>– Orphan: *Cd97, Celsr2, Darc, Eltd1, Emr1, Emr4, Gpr107, Gpr108, Gpr111, Gpr114, Gpr116, Gpr124, Gpr125, Gpr126, Gpr132, Gpr133, Gpr137, Gpr137b, ~~Gpr146~~, Gpr153, ~~Gpr171~~, ~~Gpr174~~, Gpr176, Gpr18, ~~Gpr182~~, Gpr183, Gpr19, Gpr21, Gpr30, Gpr34, ~~Gpr35~~ Gpr39, Gpr4, ~~Gpr52~~, Gpr56, Gpr63, Gpr64, Gpr65, Gpr83, Gpr97, Gprc5a, Gprc5b, Gprc5c, Lgr4, Lgr5, Lgr6, Lphn1, Lphn2, Lphn3, Mrgprf, ~~Mrgprh~~*<br>– Prostanoid: *Ptger1, Ptger2, Ptger3, Ptger4, Ptgfr, Ptgir, Tbxa2r*<br>– Protease: *F2r, F2rl1, F2rl2, F2rl3*<br>– Purinergic: *P2ry1, P2ry10, P2ry12, P2ry13, P2ry14, P2ry2, P2ry6* | 154 (inton-span-ning: 132) |
| Cell identity | *eGFP, Myh11* (smooth muscle), *Cdh1* (epithelial), *Cdh5* (endothelial), *Ptprc* (leukocyte), *Itgam* (myeloid), *Ly6g* (neutrophil), *Cd19* (B lymphocyte), *Cd4* (CD4 T lymphocyte), *Cd8* (CD8 T lymphocyte), *Lyve1* (lymphatic EC), *Cspg4* (pericyte), *Tnni2* (skeletal muscle) | 13 |
| Cell function | – Cell activation/differentiation: *Cd25, Cd44, H2-Ab1, Pecam1, Acta2, Edn1, Dll4, Sele, Smtn, Icam1, Vcam1, Col1a2, Col3a1*<br>– Cytokines/Growth factors: *Csf2,Csf2rb, Kdr, Pdgfb, Tgfb1, Tnf, Ifng, IL10, Il17a, Il1b, Il2, Il6*<br>– Transcription factors: *Rorc, Tbx21, Mki67, Egr1, Fos, Hif1a, Hey2, Klf2*<br>– Reference genes: *Actb, Gapdh, Hprt* | 36 |

In addition to the 122 GPCRs identified in NanoString multiplex RNA analysis, 32 receptors that were negative in NanoString analysis were included. Also 13 genes identifying individual cell types as well as 36 function-related genes, including three reference genes, were added. Whenever possible an intron-spanning design was used, but for 22 single exon GPCRs this was not possible (indicated by strikethrough).

identifying individual cell types and 36 function-defining genes, including the three reference genes β-actin (*Actb*), glyceraldehyde 3-phosphate dehydrogenase (*Gapdh*) and hypoxanthine phosphoribosyltransferase (*Hprt*), were included in the array (Table 1). Whenever possible intron-spanning primer design was used, but for 22 single exon GPCRs this was not possible. These 22 non-intron-spanning primer pairs gave positive results in all tested cell types (Supplementary Fig. 1B,C), indicative of contamination with genomic DNA. GPCRs not allowing intron-spanning primer design were therefore excluded from the analysis (strikethrough in Table 1). Individual EC or SMC were pre-sorted by flow cytometry from enzymatically digested tissues of mice expressing enhanced green fluorescent protein (EGFP) in EC or SMC and subjected to single-cell isolation, cDNA synthesis and RT-PCR amplification using a microfluidic-based system. Various vascular beds were analysed, including aorta (ao), skeletal muscle vasculature (sk), lung (lu) or brain (br). All primer pairs included in the array produced amplification products with the predicted melting behaviour in at least one of three cell types tested (EC, SMC and leukocytes) (Supplementary Table 3). Despite stringent sorting criteria, single-cell analysis of individual EC or SMC showed 10–18% contamination with other cell types (Table 2). Only cells with clear identity ('*Cdh5* only', '*Myh11* only') and positive expression of reference genes *Gapdh* and *Hprt* as quality control were included in further analyses (Fig. 1a). A systematic comparison of expression data obtained by bulk RNA analysis or single-cell RT-PCR in SMC from skeletal muscle vasculature (SMsk) showed that of 74 GPCRs undetectable in bulk RNA, 26 showed expression in individual SMsk cells (Fig. 1b, left). Of 11 GPCRs with uncertain result in bulk cDNA, all showed amplification in individual cells (Fig. 1b, middle). Of 37 GPCRs clearly expressed in bulk RNA, two were completely absent in individual cell analysis (Fig. 1b, right). These two GPCRs, *Eltd1* and *Gpr116*, are known to be highly specific for EC[23], indicating that these discrepancies are due to contamination of SMsk with EC.

**GPCR heterogeneity in SMC.** In individual aortic SMC (SMao) of healthy adult mice, GPCR expression was very heterogeneous (Fig. 2a). In total, 76 GPCRs were detected in SMao, but only 19 of them were expressed in more than 50% of cells, and only eight GPCRs (*Lphn1, Lgr6, F2r, Adra1d, Cd97, Gpr107, Gpr108* and *Mrgprf*) were expressed in more than 90% of cells (Fig. 2a). Reference genes *Actb* and *Gapdh* as well as SMC marker gene *Myh11* were homogenously expressed (Fig. 2a, top). On average, individual cells expressed $20.3 \pm 0.9$ of the tested 132 GPCRs, though the individual values varied between 3 and 38 GPCRs per cell (Fig. 2b). mRNA sequencing of individual SMao showed even lower frequencies of GPCR expression (selected data in Supplementary Fig. 2, whole data set in Supplementary Data 1), which led us to verify single-cell expression data in GPCR reporter mice that are genetically engineered to express β-galactosidase (βgal) under control of different GPCR promoters. Flow cytometric analysis of βgal expression in freshly

isolated SMao from *Mrgprf-*, *Gabbr1-* or *Gprc5b-*reporter mice closely matched the results of the single-cell RT-PCR expression analysis, while mRNA sequencing data underestimated GPCR expression (Fig. 2c,d). We also sequenced single-cell RT-PCR amplicons to exclude off-target amplification or amplification of highly homologous GPCRs and found that the amplified sequences were in all cases specific for the targeted receptor (Supplementary Fig. 3 and Supplementary Data 2). To investigate whether the same degree of heterogeneity was present on the protein level, we analysed expression of select GPCRs in individual SMao by flow cytometry. We found that also on the protein level GPCR expression was heterogeneous, and that the percentages roughly matched the values obtained by single-cell RT-PCR (Supplementary Fig. 4A,B). *Ex vivo* culture of primary SMao (passage 1) resulted not only in an upregulation of genes indicative of a dedifferentiated SMC phenotype, for example, intercellular adhesion molecule 1 (*Icam1*), vascular cell adhesion molecule 1 (*Vcam1*), marker of proliferation Ki-67 (*Mki67*) or interleukin-6 (*Il6*), but also in strongly increased expression frequency for the majority of GPCRs (Fig. 2e). Some GPCRs also showed decreased expression, such as *Lgr6*, *Npy1r*, *Crhr2* or *Bdkrb2* (Fig. 2e), but the average number of GPCRs per cell was still strongly increased (Fig. 2f). GPCR expression frequencies in cultured human aortal SMC (passage 1) matched in most cases the murine data (Fig. 2g middle), though some GPCRs showed notable difference between the species or between individual human donors (Fig. 2g right).

**GPCR repertoire in different types of SMC.** We next investigated whether the strong heterogeneity of GPCR expression was a special feature of conductance arteries such as the aorta, or whether it was also present in resistance arteries, for example, from SMsk. Also in SMsk GPCR expression was very heterogeneous (Fig. 3a), although the overall number of GPCRs per cell was higher than in SMao (Fig. 3a,b). The repertoire of GPCRs expressed in the two SMC types differed strongly: GPCRs *Lphn2, Cmklr1, Lpar1, Gpr133, P2ry6, Lgr6* and *F2r* were mainly present in SMao, while a large number of other GPCRs were predominantly expressed in SMsk (Fig. 3a). Among those receptors preferentially or selectively expressed in SMsk, the largest group were peptide hormone receptors (Fig. 3c), for example, receptors for endothelin (*Ednra, Ednrb*), angiotensin II (*Agtr1a*), vasopressin (*Avpr1a*), neuropeptide Y (*Npy1r*), pituitary adenylate cyclase-activating polypeptide (PACAP, acting on *Adyap1r1*), vasoactive intestinal polypeptide (VIP, acting on *Vipr1* and *Vipr2*), corticotropin releasing hormone (*Crhr2*), parathyroid hormone (*Pth1r*) or calcitonin gene-related peptide (CGRP, acting on *Calcrl*) (Fig. 3d). Also a number of chemokine and orphan receptors were preferentially expressed in SMsk (Fig. 3e). We also analysed GPCR expression in other SMC types, for example, from the mesenteric vasculature (SMmes) or urinary bladder (SMub). K-means cluster analysis revealed that SMC from mesenteric or skeletal muscle vasculature, two regions rich

**Table 2 | Contaminating cells in single-cell expression analysis.**

| Cell type | Sorted for | Marker gene expression (%) | | | |
|---|---|---|---|---|---|
| | | Cdh5 only | Myh11 only | Other markers | No marker |
| ECsk | Cdh5-EGFP | 81.5 | 0.0 | 17.8 | 0.7 |
| EClu | Cdh5-EGFP | 80.8 | 0.0 | 11.6 | 7.6 |
| ECbr | Cdh5-EGFP | 85.4 | 0.0 | 10.4 | 4.2 |
| SMao | Myh11-EGFP | 0.0 | 81.5 | 15.2 | 3.2 |
| SMsk | Myh11-EGFP | 0.8 | 84.0 | 12.0 | 3.2 |

Sorted EC and SMC were subjected to single-cell expression analysis and re-evaluated based on the expression of various identity-defining genes.

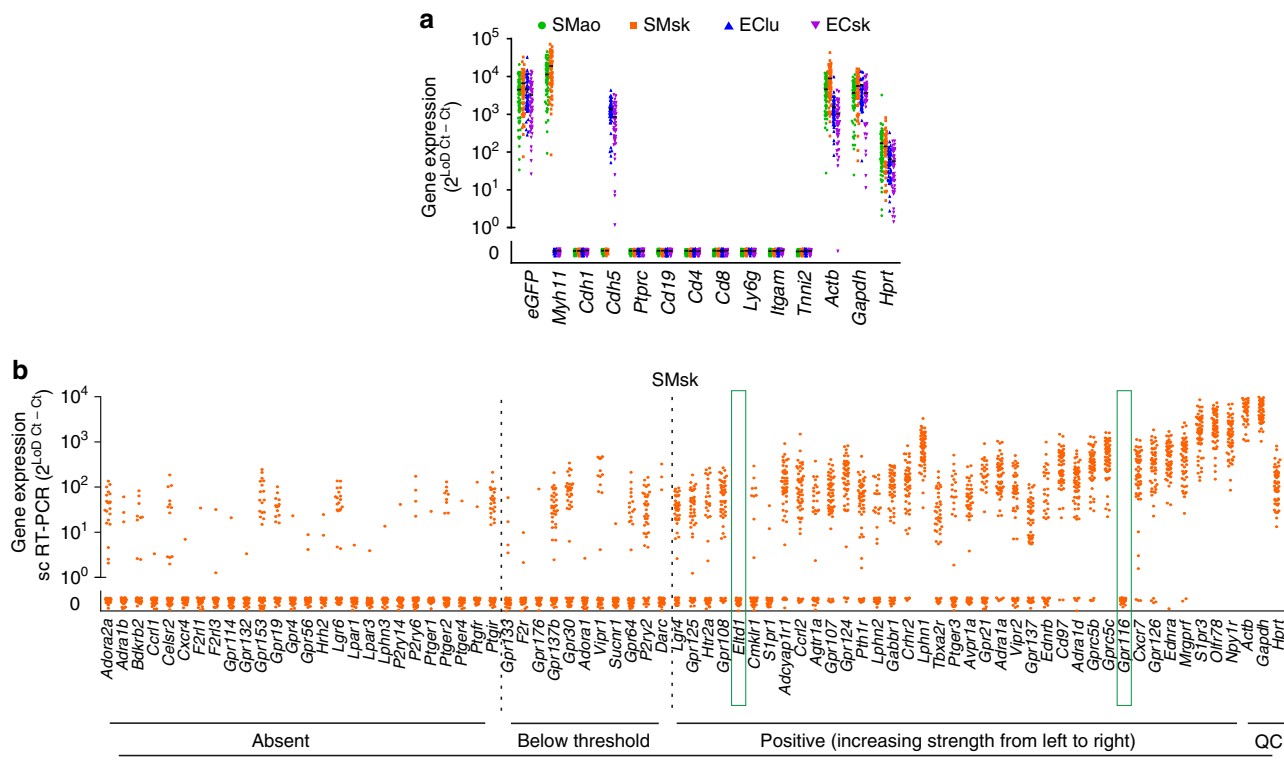

**Figure 1 | Single-cell RT-PCR analysis in freshly isolated vascular cells.** EC or SMC obtained from skeletal muscle vasculature (sk), lung (lu), aorta (ao) of Cdh5-Crepos; dTom/EGFP-reporterpos mice (Cdh5-EGFP) and tamoxifen-treated Myh11-CreERT2pos; dTom/EGFP-reporterpos mice (Myh11-EGFP), respectively, were subjected to single-cell RT-PCR. (**a**) Expression of identity-defining genes and quality control genes (*Gapdh, Hprt* and *Actb*) after exclusion of contaminating cells or marker negative cells (each dot one cell). (**b**) Comparison of expression data obtained by multiplex RNA expression analysis in pooled SMsk and single-cell RT-PCR (sc RT-PCR) in individual SMsk. GPCRs are arranged on the abscissa according to their expression strength in pooled RNA analysis, the ordinate shows the strength of gene expression in individual cells (each dot one cell). Green boxes indicate genes negative in sc RT-PCR but positive in pooled RNA analysis (cell/animal numbers: sc RT-PCR: $n = 57$ cells from seven mice; pooled RNA analysis: 500 ng from $10^6$ cells per six mice. Values of genes that are not expressed were scattered around 0 to allow graphical estimation of cell counts). Expression data are calculated as follows: Gene expression $= 2^{(Limit\ of\ detection(LoD)\ Ct - sample\ Ct)}$; LoD Ct is set to 24.

in resistance arteries/arterioles, were quite similar in their GPCR profile, but differed strongly from SMao or SMub (Fig. 3f,g). An analysis of differential gene expression for all SMC types is shown in Supplementary Fig. 5A–C; a comparison of GPCR frequencies for all SMC types can be found in Supplementary Fig. 5D. An overview of the total number of cells, mice and independent experiments is shown in Supplementary Table 4.

**GPCR heterogeneity in different types of EC.** To determine GPCR heterogeneity in primary murine EC from different locations, we isolated individual EC from lung, skeletal muscle or brain (EClu, ECsk, ECbr) (selected GPCRs in Fig. 4a, all data in Supplementary Fig. 6). Like SMC, EC showed a high heterogeneity of GPCR expression, with only five receptors (*Cd97, Calcrl, Gpr116, S1pr1* and *Eltd1*) being homogenously expressed in all EC types (Fig. 4a, Supplementary Fig. 6). Individual EC from different vascular beds expressed on average 20.9, 21.0 and 16.3 of the tested 132 GPCRs (Fig. 4b). K-means cluster analysis identified three-cell clusters, which largely corresponded to the three different EC types (Fig. 4c) and are characterized by specific GPCR repertoires: receptors *Glp1r, Calcrl, Lphn3, Ccbp2, Celsr2* and *Cd97* were strongest expressed in cluster 1 cells (containing EClu) (Fig. 4d), whereas receptors *Darc, Ptger4, P2ry6, Cysltr1* or *Gprc5a* were strongest in cluster 2 cells (containing mainly ECsk) (Fig. 4e). Cluster 3 cells (mainly ECbr) showed higher expression of *Gpr30, Gpr124, Gpr4* or *Lpar4* (Fig. 4f).

**Endothelial GPCR pattern after acute inflammatory activation.** We next investigated how the GPCR repertoire changes in individual EC in response to acute inflammatory activation. To do so, we used a murine sepsis model induced by intraperitoneal (i.p.) injection of bacterial endotoxin lipopolysaccharide (LPS), which results in massive direct and indirect activation of EC[24]. ECbr showed 12 h after LPS injection a clear upregulation of markers of inflammatory activation such as *Icam1* or E-selectin (*Sele*), whereas expression of vascular endothelial growth factor receptor 2 (VEGFR2, encoded by *Kdr*) or endothelin-1 (*Edn1*) was reduced (Fig. 5a). Also the GPCR pattern changed significantly: numerous orphan receptors (*Gprc5a, Gpr97, Gpr111, Gpr153, Lphn2, Gpr107, Gprc5b* and *Gpr56*) were upregulated, as well as select chemokine receptors (*Darc, Ccrl2* and *Cxcr7*) and purinergic receptors (*P2ry2, P2ry6* and *Adora2a*). Other GPCRs, such as *Tbxa2r, Lphn1, Gpr125, Gabbr1, Gpr124, Calcrl* or *Cd97*, were downregulated (Fig. 5a), resulting in total in a non-significant increase in the number of GPCRs per individual cell (Fig. 5b). K-means cluster analysis confirmed ECbrLPS as a distinct population characterized by a specific GPCR repertoire (Fig. 5c,d). Also in EClu characteristic changes in the GPCR expression pattern were observed upon LPS treatment, though with distinct differences to ECbr (select data in Fig. 5e,f; all data in Supplementary Fig. 6); the average GPCR number per cell was clearly reduced ECluLPS (Fig. 5g). K-means analysis of all four groups showed that upon LPS activation the originally very dissimilar EC types became more similar, but are still recognized

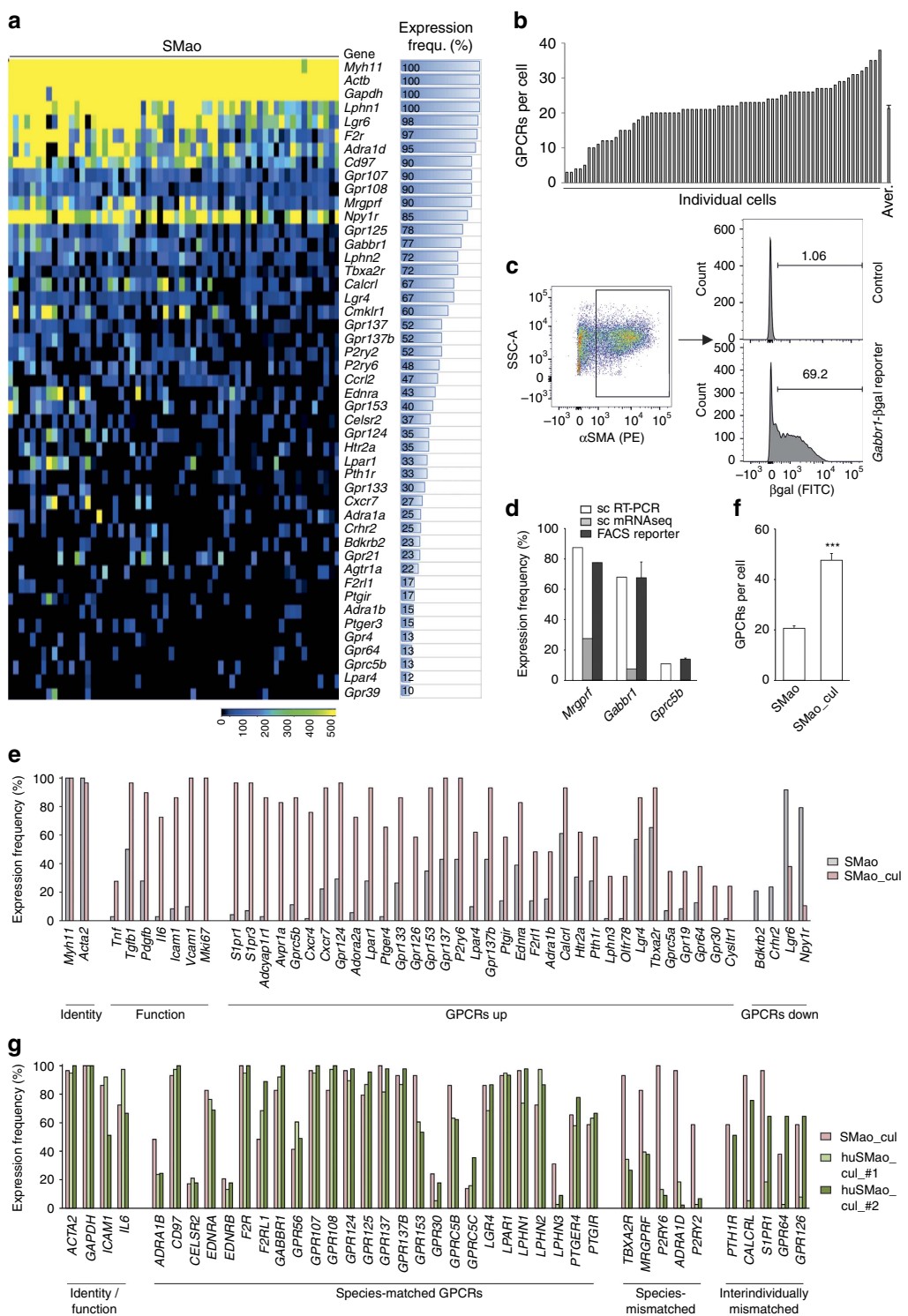

**Figure 2 | GPCR expression in individual aortal SMC (SMao).** (**a**) Heat map of GPCR expression in 60 SMao (each column one cell) from eight mice (sorted by frequency; horizontal bars on the right side visualize expression frequency (in %); frequencies <10% not shown (for full data set, Supplementary Fig. 6); expression of *Myh11*, *Actb* and *Gapdh* as quality control). (**b**) Number of GPCRs expressed in individual SMao. (**c**) Example of flow cytometric analysis of βgal-positive SMao in control mice and *Gabbr1*-βgal reporter mice. (**d**) Comparison of GPCR expression frequency in individual cells as judged by single-cell RT-PCR (sc RT-PCR), single-cell mRNA sequencing (sc mRNAseq), and flow cytometric analysis of reporter-positive SMao from *Mrgprf-/Gabbr1-/Gprc5b*-reporter mice (FACS reporter, n = 3 per group). (**e**) Percentage of cells expressing selected genes in freshly isolated SMao (n = 60) and SMao cultured for one passage (7–10 days) (SMao_cul; n = 29). (**f**) Average number of GPCRs expressed in freshly isolated SMao and SMao_cul. (**g**) Percentage of cells expressing a given gene in cultured murine (SMao_cul) and human (huSMao_cul) aortal SMC (#1: 2-year-old healthy male child, n = 42 cells; #2: 51-year-old male adult, n = 31 cells); Expression data are calculated as follows: Gene expression = 2$^{(\text{Limit of detection(LoD) Ct} - \text{sample Ct})}$; LoD Ct was set to 24. Data in **b,d,f** are shown as mean ± s.e.m.; comparisons between groups were performed using two-tailed *t*-test; ***P < 0.001.

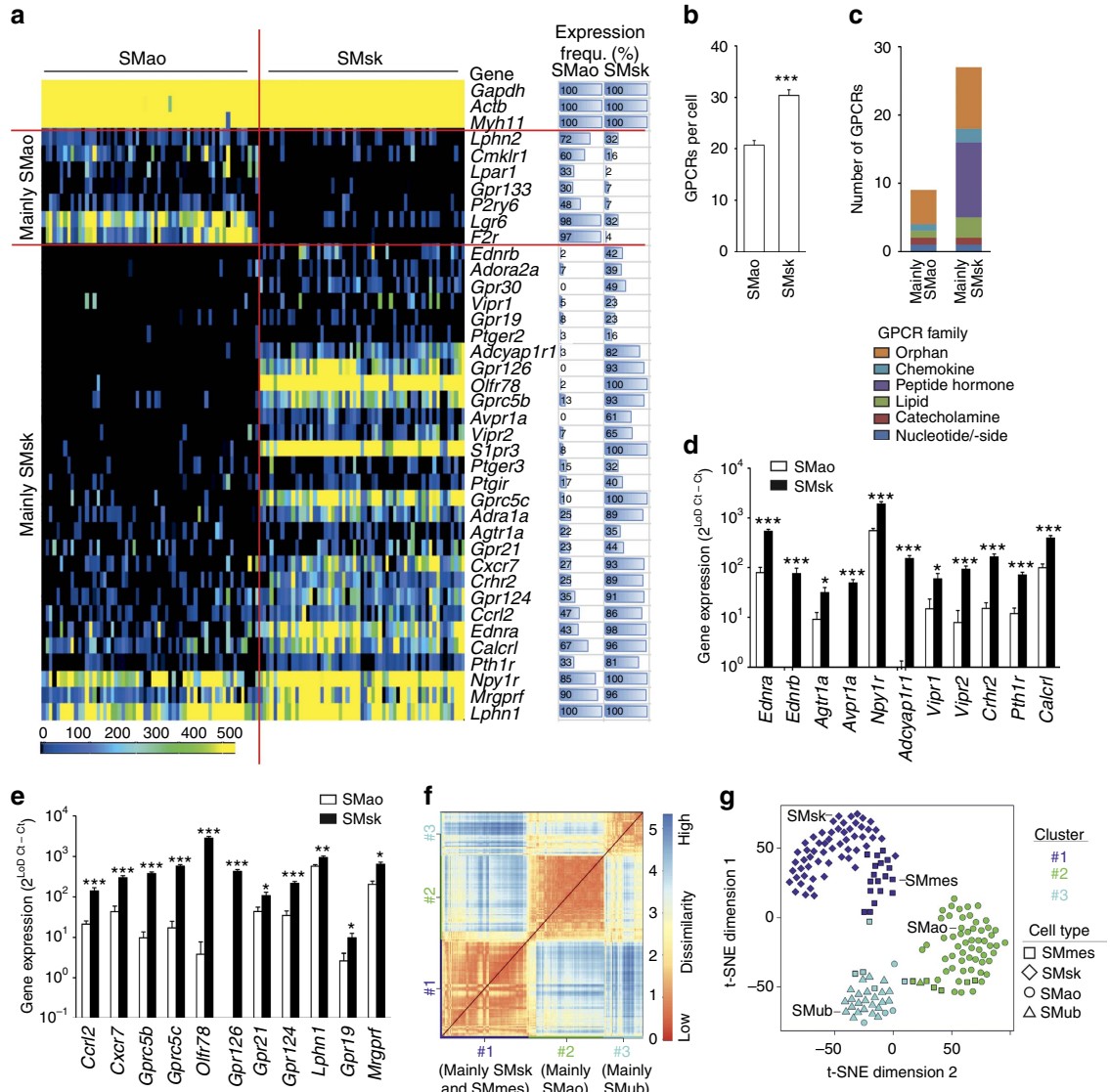

**Figure 3 | Single-cell GPCR expression in different SMC types.** (**a**) Heat map of GPCR expression in SMao and SMsk (60 and 57 cells per SMC type, from seven to eight mice each); horizontal bars on the right side visualize expression frequency (in %) (for full data sets, Supplementary Fig. 6). (**b**) Average number of GPCRs expressed in individual SMao and SMsk. (**c**) GPCRs preferentially expressed in SMao or SMsk grouped by GPCR family. (**d,e**) Comparative analysis of expression strength for different peptide hormone receptors (**d**) or members of the chemokine/orphan family (**e**) in SMao and SMsk. (**f,g**) K-means clustering results for different SMC types: (**f**) Heat map indicating degree of similarity between individual cells as measured by Euclidian distances of the transcriptome correlation matrix. K-means clustering identified three major groups of cells which are colour-coded along the axes. (**g**) t-SNE map representation of clusters identified in **f**: the more similar two cells are, the closer together they are plotted (each dot one cell). Cluster assignment is indicated by colour, cell type is indicated by symbols. SMmes, SMC from mesenteric vasculature (29 cells from eight mice); SMub, SMC from urinary bladder (25 cells from eight mice). Expression data are calculated as $2^{(Limit of detection(LoD) Ct-sample Ct)}$; LoD Ct was set to 24. Data in **b,d,e** are shown as mean ± s.e.m.; comparisons between groups were performed using two-tailed $t$-test. *$P < 0.05$; **$P < 0.01$; ***$P < 0.001$.

as separate clusters (Fig. 5h). Detailed analysis of transcriptional changes showed that both EC types showed upregulation of *Icam1, Vcam1* and *Sele*, whereas *Kdr,* platelet-derived growth factor b (*Pdgfb*), and *Edn1* were downregulated (Supplementary Fig. 7A). Both EC types showed upregulation of GPCRs *Ccrl2, Cxcr7, Gpr111* and others (Supplementary Fig. 7B), whereas *Calcrl, Cd97, Gabbr1, Lphn1* and so on were downregulated (Supplementary Fig. 7C). However, some GPCRs were differentially regulated: *Ptger2, P2ry10, F2rl1, Ccbp2, Glp1r, Celsr2* or *Cxcr4* were up- or downregulated selectively in EClu (Fig. 5i), while *Gpr107, Gpr97, Adora2a, Gpr153, Darc, Fpr2, C5ar1* and *Ccr1* were only upregulated in ECbr (Fig. 5j). Interestingly, upregulation of *Fpr2, C5ar1* and *Ccr1* was restricted to a small subpopulation of ECbrLPS that showed at

the same time reduced expression of GPCRs such as *Darc, Gpr97, Gabbr1* and others (Fig. 5k). Though expression of *Fpr2, C5ar1* and *Ccr1* is normally restricted to myeloid cells (Supplementary Fig. 6), this subpopulation was clearly positive for *Cdh5, Pecam1* and EC-specific GPCRs such as *Gpr116* or *Eltd1,* but negative for leukocyte markers *Ptprc, Itgam* and *Ly6g* (Fig. 5k), indicating they are indeed an EC population that assumed a myeloid-like GPCR expression pattern.

To also assess the endothelial GPCR repertoire under conditions of chronic vascular inflammation, we investigated aortal EC from ApoE-deficient mice kept for 16 weeks on a Western-style high-fat diet[25]. Surprisingly, all *Cdh5*-positive EC obtained from aortae were also positive for SM marker *Myh11,* although *Myh11* levels were very low compared to normal SMC

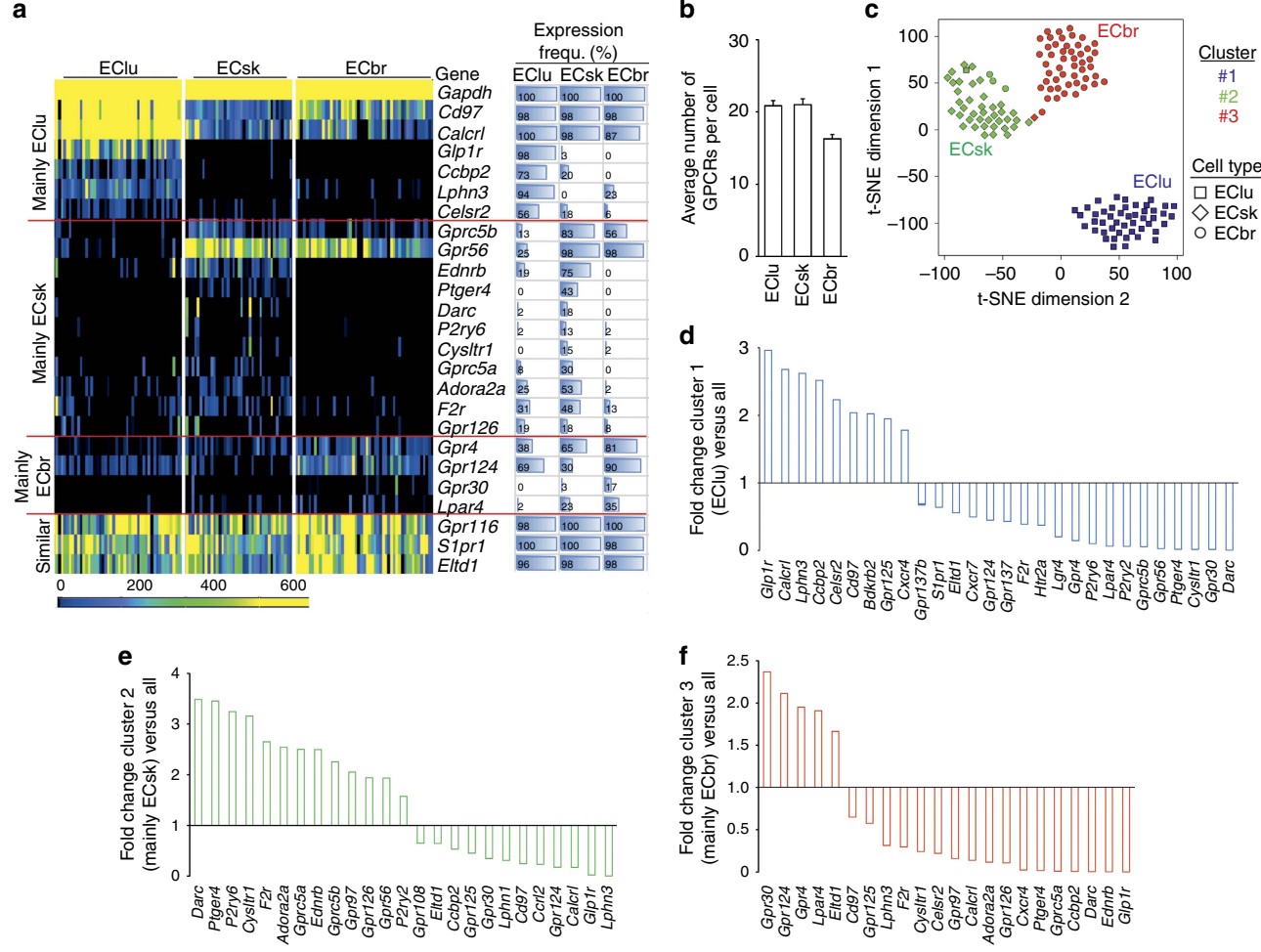

**Figure 4 | Single-cell GPCR expression in different EC types.** (**a**) Heat map of GPCR expression in EClu, ECsk, ECbr (48, 40, 52 cells per EC type, from six to eight mice per group); horizontal bars on the right side visualize expression frequency in % (for full data sets, Supplementary Fig. 6). (**b**) Average number of GPCRs expressed in individual EC types (mean ± s.e.m.). (**c**) T-SNE plot of k-means clustering data for different EC types: cluster assignment is indicated by coloured numbers, cell type is indicated by symbol (each dot one cell; distance between dots indicates degree of similarity). (**d**–**f**) Genes differentially expressed in cluster 1 (EClu, **d**), cluster 2 (mainly ECsk, **e**), and cluster 3 (mainly ECbr, **f**) (only genes significantly regulated ($P > 0.05$) and with fold change $< 0.7$ or $> 1.5$ are displayed). Data in **b** are mean ± s.e.m.

(Fig. 6a). *Cdh5*^pos; *Myh11*^low aortic cells expressed EC-specific GPCRs such as *Gpr116* or *Eltd1*, whereas GPCRs normally present in SMao, such as *Lgr6* or *Mrgprf*, were low or absent (Fig. 6b). We therefore concluded that these *Cdh5*-positive cells are despite low levels of *Myh11* expression aortic EC (ECao). Compared to cells from healthy aortae, ECao from atherosclerotic aortae showed clear changes in the expression of both function-related genes and GPCRs (Fig. 6c). In detail, markers of inflammatory activation such as *Icam1*, *Vcam1* or *Il6* were upregulated, whereas expression of *Kdr* and *Edn1* was reduced (Fig. 6c,d). Most GPCRs showed diminished expression (Fig. 6c,d), resulting in a reduction of the GPCR number expressed per cell (Fig. 6e). However, a trend to increased gene expression was observed for *Ccrl2*, *S1pr3*, *Darc* and *Gpr153* (Fig. 6c,d). A comparison of expression changes in acutely and chronically activated EC showed that adhesion molecules and selected GPCRs such as *Ccrl2*, *Darc*, *F2r*, *Gpr153* or *Lphn2* were upregulated both in acute and chronic inflammation, whereas *Kdr*, *Edn1*, *Pdgfb* and various GPCRs were downregulated in both conditions (Fig. 6f). Interestingly, a number of GPCRs were upregulated in acute inflammatory activation, but downregulated in chronic inflammation, for example, *Cxcr7*, *Gpr56*, *Gprc5a* and others (Fig. 6f).

**GPCR repertoire in SMC from atherosclerotic mice.** We next studied GPCR patterns in SMC from atherosclerotic aortae. After 16 weeks of high-fat diet, SMao showed clear signs of inflammatory activation and dedifferentiation[26,27]: *Icam1*, *Vcam1* and *Il6* were upregulated, whereas contractile proteins such as *Acta2* and *Myh11* were reduced in expression strength, though not in frequency (Fig. 7a, upper part). In addition, numerous GPCRs showed increased expression frequency (Fig. 7a, lower part), resulting in a significantly increased number of GPCRs per individual cell (Fig. 7b). K-means cluster analysis assigned the majority of SMao from atherosclerotic mice (squares in Fig. 7c) to a cluster characterized by increased expression of GPCRs such as *Olfr78*, *Ednrb*, *Adcyap1r1*, *Ptger2*, *Avpra1a*, *S1pr3*, *Ptgir*, *Vipr2*, *Ccrl2*, *Cxcr7*, *Gprc5b* or *Gprc5c* (Fig. 7d). ApoE-deficient mice without high-fat diet showed an intermediate phenotype, but no clear upregulation of inflammatory genes such as *Icam1* or *Vcam1*; comparable changes were observed in aged but otherwise healthy C57BL/6 mice (Fig. 7e). To understand the mechanisms regulating GPCR expression in healthy and dedifferentiating SMao, we analysed transcription factor (TF) binding sites in promoters of GPCRs that were either up- or downregulated in dedifferentiating SMC. GPCRs that were upregulated in atherosclerotic SMao were more likely to contain binding sites

for TFs such as heat shock factors proteins 1,2,4 (HSF1,2,4), retinoic acid receptor α/β (RARA, RARB), NF-κB2, AP2 (TFAP2A,B,C), KLF5 and others (Fig. 7f, left side). GPCRs that were downregulated in atherosclerotic SMao were more likely to

contain bindings sites for, among others, estrogen-related nuclear receptors ESRRA and ESRRB, sterol regulatory element-binding transcription factor 1 (SREBF1), or T box TFs TBX1 and TBX19 (Fig. 7f, right side; complete list in Supplementary Table 5).

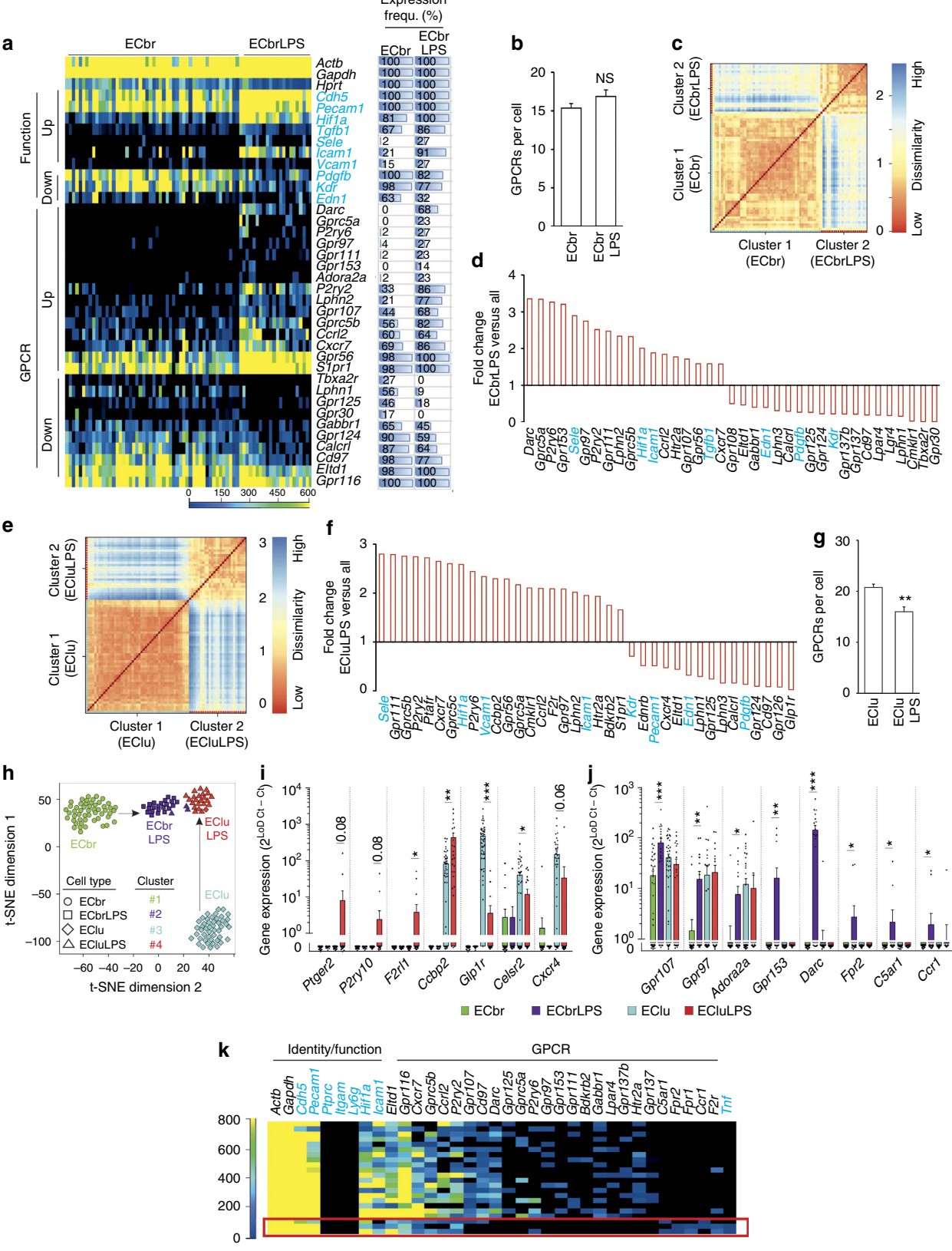

**Dedifferentiating SMC of the healthy aorta.** We noticed that some SMao from healthy mice were assigned to the cluster of dedifferentiating SMC in Fig. 7c (marked by arrows), indicating that a spontaneously dedifferentiating population exists in the healthy aorta. Indeed, k-means analysis of SMao from healthy mice identified a small subpopulation of cells as being clearly different from the rest (cluster 3 in Fig. 8a). These cells were characterized by a reduced expression of contractile proteins *Myh11* or *Acta2*, but increased expression of *Icam1*, *Vcam1*, *Col1a2* and *Col3a1* (Fig. 8b). This dedifferentiated phenotype was associated with reduced expression of GPCRs such as *Lgr6* and *Adra1d*, and increased expression of *Ptgir*, *Vipr2*, *Gpr39*, *Lpar1*, *Gprc5b*, *Gpr124*, *Cxcr7*, *Gpr137b*, *Ednra* and others (Fig. 8b). Spearman's rank correlation analysis confirmed the positive correlation between *Myh11*, *Acta2*, *Lgr6*, *Adra1d* on the one hand and *Icam1*, *Vcam1*, *CD44*, *Ptgir*, *Gpr39*, *Vipr2*, *Cxcr7*, *Gprc5b* and so on on the other hand (Fig. 8c). A direct comparison of changes in dedifferentiating SMC from healthy mice and atherosclerotic mice showed that both cell types behaved similarly with respect to upregulation of adhesion molecules and GPCRs such as *Ptgir*, *Vipr2*, *Gprc5b*, *Gprc5c*, *Cxcr7* and so on, as well as down-regulation of *Acta2*, *Myh11*, *Adra1d* and *Lgr6* (Fig. 8d, left side). However, a number of GPCRs were upregulated only in the context of atherosclerotic dedifferentiation, and not in spontaneous dedifferentiation, such as *Olfr78*, *Ednrb*, *Adcyap1r*, *Ptgfr* and others (Fig. 8d, right side).

Since vascular SMC dedifferentiation is believed to occur mainly in regions of disturbed flow, for example, the inner curvature of the aortic arch[28], we used *Gprc5b*-βgal reporter mice to investigate the localization of spontaneously dedifferentiating SMC *in vivo*. Immunohistochemical analysis of the expression of βgal and SMC marker α-smooth muscle actin (αSMA) in transverse section of the aortic arch (Fig. 8e) showed only sparse βgal expression in SMC of the outer curvature, but an enrichment of βgal/αSMA-positive cells in the inner curvature (Fig. 8f,g). These data confirm the notion that GPCRs such as *Gprc5b* are selectively expressed in SMC in atheroprone regions. We finally investigated whether selective targeting of GPCRs that are specifically expressed in dedifferentiating SMao may be used to modulate their functional state. Knockdown of *Gprc5b* resulted in freshly isolated SMao from ApoE-deficient mice in a significant increase in expression of *Icam1* and *Il6*, indicating that this receptor plays a modulatory role in inflammatory gene expression in dedifferentiating SMC (Fig. 8h).

## Discussion

We report here the analysis of GPCR expression in primary vascular cells on the single-cell level. Compared to conventional expression analysis in pooled cell RNA/cDNA, single-cell expression analysis has three crucial advantages: First, it allows to rigorously exclude contaminating cells and thereby precludes misinterpretation of data. Second, single-cell expression analysis is able to detect transcripts in rare cell populations that might be below threshold in bulk RNA/cDNA analyses: 37 GPCRs that were judged negative or uncertain in SMsk by NanoString analysis were detected in individual SMsk, though in some cases only in 1.5% of cells. However, due to the limited amount of RNA obtained from an individual cell also single-cell expression may overlook very low abundance transcripts, in particular if the RNA quality is compromised. Since GPCRs are in many cases expressed at low levels[29], we directly compared the two major readout systems for single-cell expression analysis, RT-PCR and mRNA sequencing, with respect to their sensitivity for GPCR detection in SMao. RT-PCR detected most GPCRs with higher frequency than mRNA sequencing, and sequencing of the single-cell amplicates as well as analyses in GPCR reporter mice and on the protein level largely confirmed the RT-PCR data. A possible explanation for this difference in sensitivity lies in the fact that single-cell RT-PCR uses target-specific pre-amplification of mRNAs, while pre-amplification for mRNA sequencing is unbiased.

The third and probably most relevant advantage of single-cell expression analysis is that it allows to estimate the degree of GPCR heterogeneity within a cell population and, consecutively, to identify correlations between GPCR profile and functional state of a given cell. We found that all types of SMC and EC showed a surprisingly high heterogeneity of GPCR expression, and reporter analysis, as well as studies on the protein level confirmed these findings. Studies in other fields, mainly developmental biology, suggest that this is not a specialty of GPCRs, but is also observed in other gene families[17,18]. These findings have major implications for pharmacotherapy, since current interpretations of GPCR expression data rely on the assumption that all cells of a given population are equal, or at least comparable[15,16,30,31]. Our data not only clearly disprove this assumption for the majority of GPCRs, they also open up the possibility to selectively target pathologically altered cells based on their specific GPCR repertoire.

We show, for example, that dedifferentiated SMao differ from normal SMao not only in the expression of typical markers indicating inflammatory activation and dedifferentiation, but also in their GPCR repertoire. Among those receptors that are preferentially expressed in dedifferentiating SMao are a number of $G_s$-coupled receptors with known anti-inflammatory and anti-proliferative properties, such as the prostacyclin receptor *Ptgir* or the VIP receptor *Vipr2* (refs 32,33). It is tempting to speculate that also other $G_s$-coupled receptors upregulated in dedifferentiating SMao will exert anti-proliferative effects, for example, the PACAP receptor *Adcyap1r1*, or the adenosine

---

**Figure 5 | Endothelial GPCR pattern after acute inflammatory activation by LPS *in vivo*.** (**a–d**) Analyses in brain EC (ECbr): (**a**) Heat map of GPCR expression in ECbr from healthy mice and LPS-treated mice (ECbrLPS) (52 and 22 cells from seven and four mice, respectively). Horizontal bars on the right side visualize expression frequency (in %) (for full data set, Supplementary Fig. 6); function-defining genes are shown in blue. (**b**) Average number of GPCRs expressed in individual ECbr from healthy or LPS-treated mice. (**c**) Heat map indicating dissimilarities between individual ECbr. K-means clustering identified two cell clusters that are colour-coded along the axes and correspond to ECbr from healthy and LPS-treated mice, respectively. (**d**) Fold difference in gene expression in ECbrLPS compared to all cells. (**e–g**) Analyses in lung EC (EClu) (48 and 25 cells from eight and four mice, respectively): (**e**) Heat map indicating dissimilarities between individual EClu. K-means clustering identified two cell clusters that are colour-coded along the axes and correspond to EClu from healthy and LPS-treated mice, respectively. (**f**) Fold difference in gene expression in ECluLPS compared to all cells. (**g**) Average number of GPCRs expressed in individual EClu from healthy or LPS-treated mice. (**h–j**) Comparison of LPS effects in EClu and ECbr: (**h**) T-SNE plot of k-means clustering data for different EC types with and without LPS treatment: cluster assignment is indicated by coloured numbers, cell type is indicated by symbol (each dot one cell; distance between dots indicates degree of similarity). (**i,j**) Comparative analysis of expression strength of selected GPCRs in different EC types. (**k**) Rearranged and extended heat map of ECbrLPS shown in **a**: *Fpr1/Fpr2/Ccr1/C5ar1*-expressing cells are indicated by red box. All expression data are calculated as $2^{(Limit\ of\ detection(LoD)\ Ct—sample\ Ct)}$; LoD Ct was set to 24. Function-defining genes are shown in blue. Data in **b,g,i,j** are means ± s.e.m.; comparisons were made using two-sample *t*-test. *$P < 0.05$; **$P < 0.01$; ***$P < 0.001$.

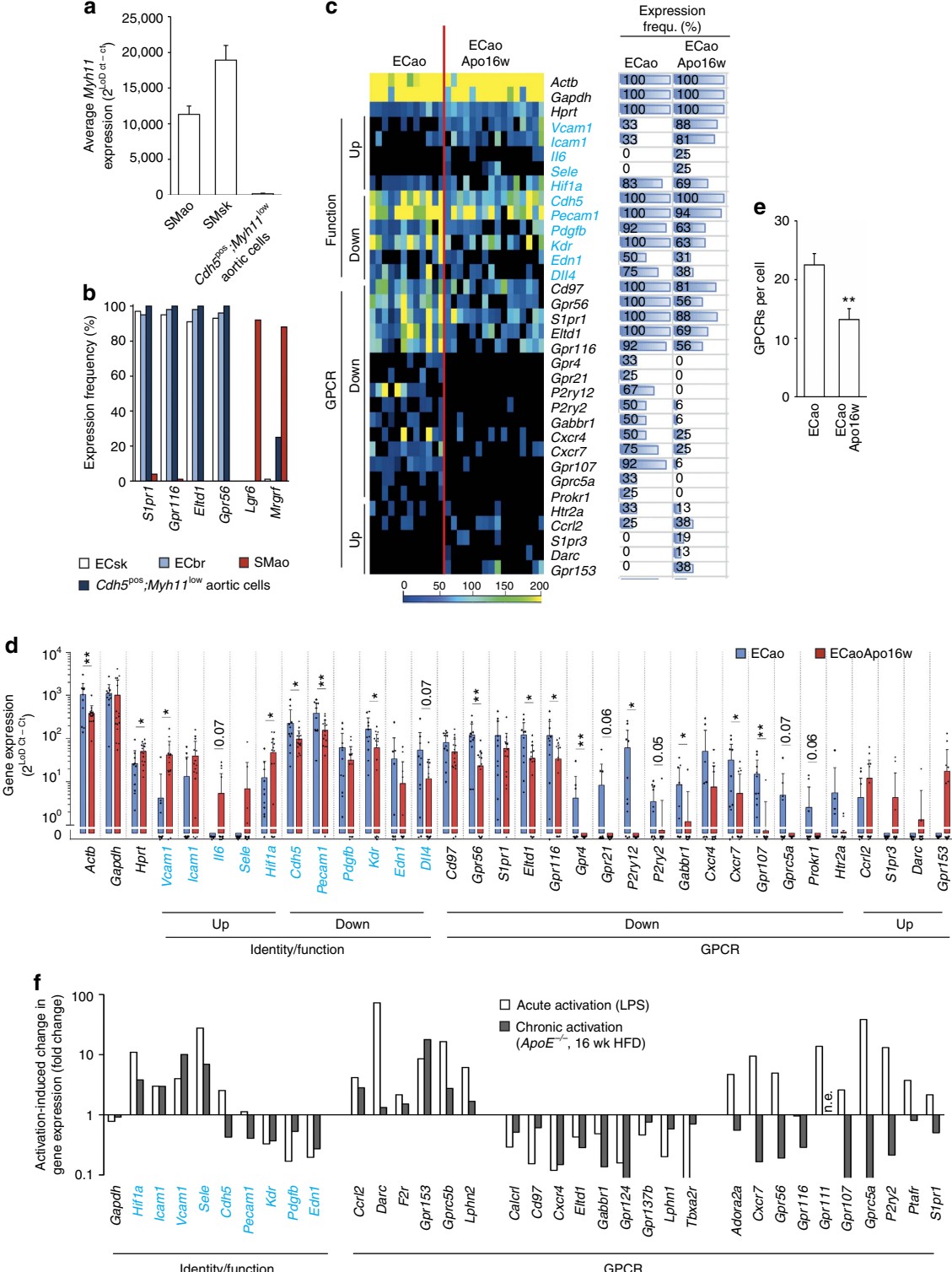

**Figure 6 | Endothelial GPCR pattern in atherosclerosis.** (**a**) Average expression strength of SM marker *Myh11* in SMC from aorta (SMao) or skeletal muscle vasculature (SMsk), or in *Cdh5*-positive aortal cells. (**b**) Percentage of cells expressing selected GPCRs in freshly isolated EC from skeletal muscle (ECsk) or brain (ECbr), as well as in *Cdh5*$^{pos}$; *Myh11*$^{low}$ aortal cells or aortal SMC (SMao). (**c**) Heat map of GPCR expression in aortal EC from healthy mice (ECao) and ApoE-deficient mice kept for 16 weeks on high-fat diet (ECaoApo16 w) (12 and 16 cells from four to six mice, respectively). Horizontal bars on the right side visualize expression frequency (in %) (for full data set, Supplementary Fig. 6). (**d**) Comparative analysis of gene expression strength in ECao and ECaoApo16w. (**e**) Average number of GPCRs expressed in individual ECao from healthy and atherosclerotic mice. (**f**) Changes in endothelial gene expression in response to acute inflammatory activation by LPS or chronic inflammatory activation in atherosclerotic mice (n.e., not expressed). All expression data are calculated as $2^{(\text{Limit of detection(LoD) Ct}-\text{sample Ct})}$; LoD Ct was set to 24. Function-defining genes are shown in blue. Data in **a,d,e,** are means ± s.e.m.; comparisons in **d,e** were performed using two-sample *t*-test. *$P<0.05$; **$P<0.01$.

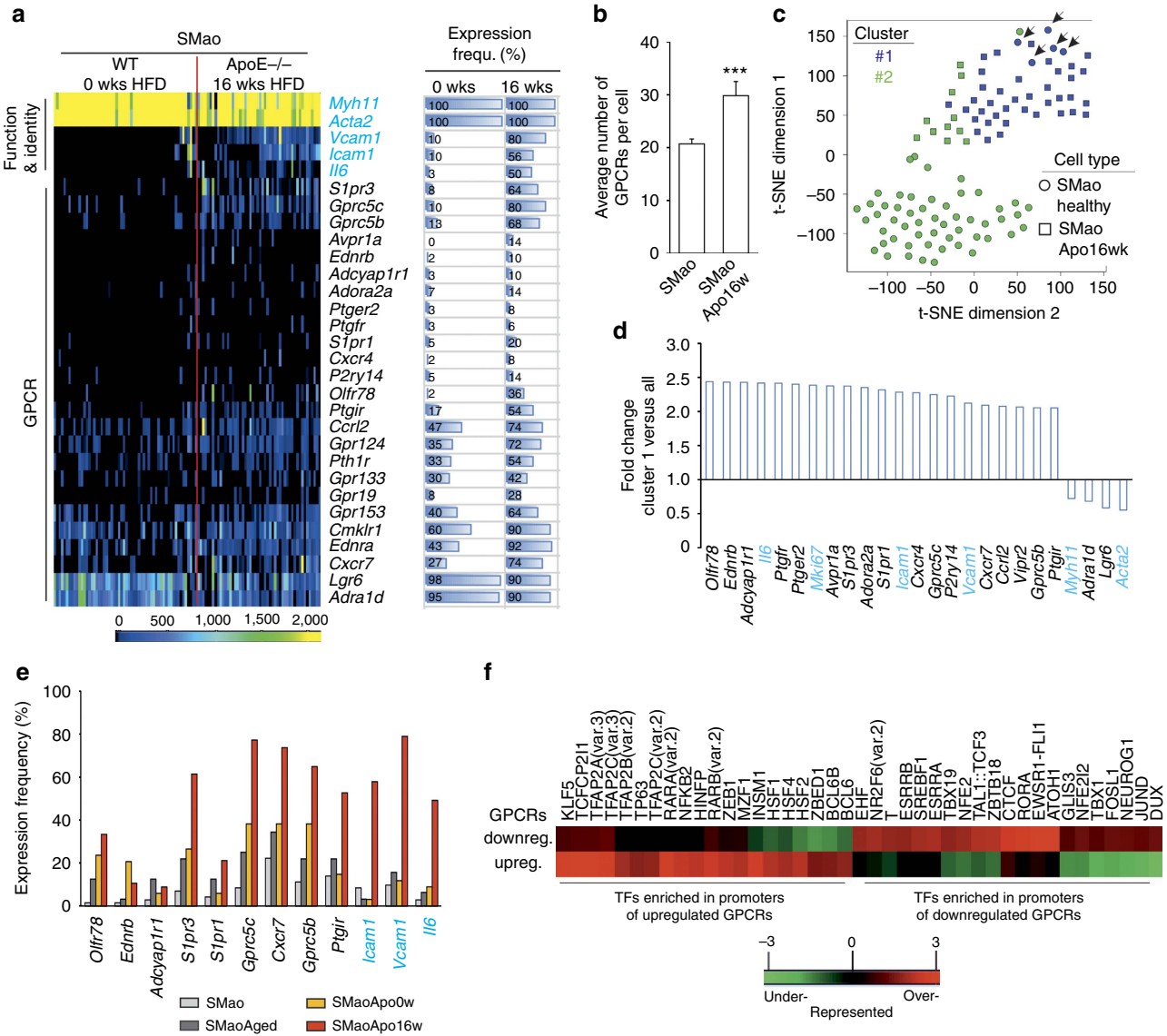

**Figure 7 | Disease-dependent changes within subgroups. (a)** Heat map of GPCR expression in SMao from healthy mice compared to ApoE-deficient mice kept for 16 weeks on high-fat diet (60 and 50 cells from eight to six mice, respectively). Horizontal bars on the right side visualize expression frequency (in %) (for full data set, Supplementary Fig. 6). **(b)** Average number of GPCRs expressed in individual SMao from healthy and atherosclerotic mice. **(c)** T-SNE representation of k-means clustering data from healthy (circles) and atherosclerotic (squares) SMao. **(d)** Genes differentially expressed in cluster 1 compared to all cells. **(e)** Expression frequency of selected GPCRs (left) and function-defining genes (right) in SMao of healthy mice aged 2–4 months (SMao, 60 cells from eight mice), healthy mice aged 16 months (SMaoAged, 32 from six mice), ApoE-deficient mice on normal diet (SMaoApo0w, 34 cells from six mice) or after 16 weeks of high-fat diet (SMaoApo16w, 50 cells from six mice). **(f)** Selection of TF binding sites significantly ($P < 0.05$) enriched (red) or depleted (green) in the promoters of GPCRs downregulated or upregulated in dedifferentiating SMao (cluster 1 from **c,d**) compared to a global promoter background set. Colour scale represents a rank based z score; for full list see Supplementary Table 5. Data in **b** are mean ± s.e.m.; comparisons were performed using two-sample t-test. ***$P < 0.001$.

receptor *Adora2a*. In line with this notion, PACAP was shown to inhibit SMC proliferation[34], and enhanced SMC proliferation was observed in Adora2a-deficient mice[35]. Furthermore, it will be particularly interesting to investigate whether those orphan receptors that are upregulated in dedifferentiating SMC, for example, *Gpr39, Gpr124, Gpr153 or Gprc5b* have the potential to positively or negatively regulate SMC differentiation. In support of this idea, we found that knockdown of Gprc5b enhanced proinflammatory gene expression in freshly isolated SMao, suggesting that this receptor negatively modulates inflammatory gene expression in SMC.

How the GPCR repertoire of an individual vascular cell is shaped and how stable it is over time, is unclear. While numerous

studies analysed the posttranslational regulation of GPCRs by phosphorylation, internalization or dimerization, their transcriptional control is little understood. To address the mechanisms regulating GPCR expression in healthy and dedifferentiating SMao, we analysed TF binding sites in promoters of GPCRs upregulated in dedifferentiating SMao. The promoters of these GPCRs were, among others, enriched in binding sites for TFs AP2, KLF5, RARA/RARB, HSF1/2/4 and NF-κB2. Some of these TFs have been implicated in SMC dedifferentiation: KLF5 has been shown to promote proliferation of vascular SMC[36], whereas activation of retinoic acid receptor α (RARA) increased migration and tissue-type plasminogen activator activity in SMC[37]. Both HSF1 and NF-κB show increased activity in

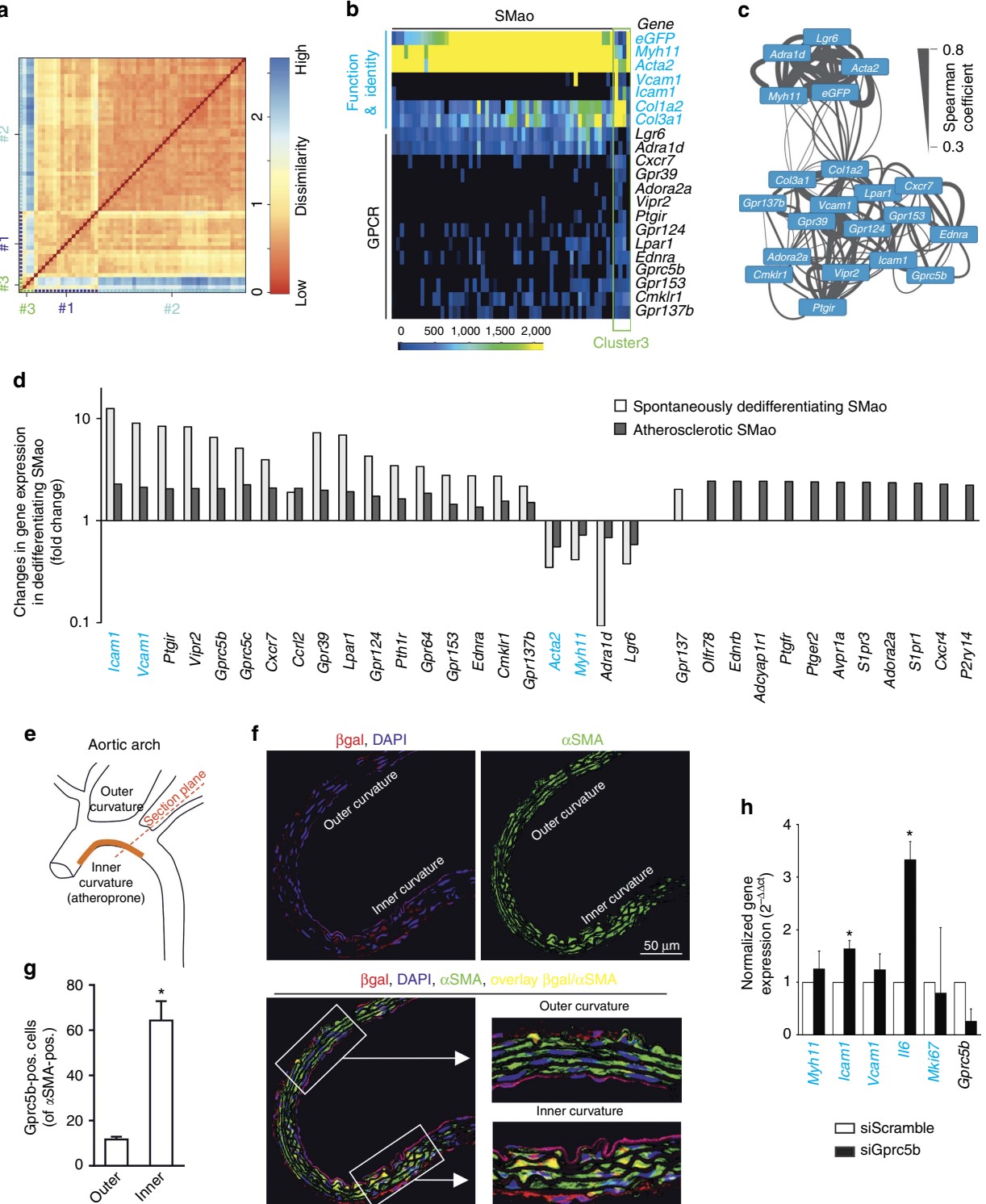

**Figure 8 | Functional subgroups within SMao.** (**a**) Heat map indicating similarities/dissimilarities between 60 individual SMao. Cell clusters identified by k-means clustering are colour-coded along the axes. (**b**) Heat map of GPCR expression in SMao (cells sorted from left to right by clusters shown in **a**). (**c**) Graphical representation of Spearman's rank correlation coefficients calculated for selected genes expressed in SMao (width of connecting line indicates strength of correlation). (**d**) Changes in gene expression in dedifferentiating SMC from healthy aortae or atherosclerotic aortae. (**e**–**g**) Immunohistochemical analysis of βgal and αSMA expression in transverse sections of the aortic arch of *Gprc5b*-βgal reporter mice (schematic diagram in **e**, exemplary photomicrographs in **f**, quantification in **g**. (**h**) Gene expression in SMao cultured for 7 days after knockdown of Gprc5b (normalized to *Actb* expression). Data in **g**,**h** are shown as mean ± s.e.m.; comparisons were performed using two-sample *t*-test (**g**) or one-sample *t*-test (**h**) ($n = 4$–$6$). *$P < 0.05$.

atherosclerotic vessels[22,38], and NF-κB activation enhances expression of inflammatory genes such as *Vcam1* and reduces expression of SM-specific genes[39]. These data suggest that TFs implicated in the regulation of SMC dedifferentiation also control changes in the GPCR repertoire, which in turn might modulate the dedifferentiation process.

Our data also show that both SMC and EC express distinct GPCR repertoires depending on the vascular bed they originate from. SMC from vascular beds rich in resistance arteries, here SMsk and SMmes, express significantly more GPCRs than aortic SMC, in particular more peptide hormone receptors and orphan receptors. Most of the peptide hormones in question have been shown to affect vascular tone: vasorelaxation is, for example, mediated by the gastrointestinal hormones VIP and PACAP (acting on receptors encoded by *Vipr1*, *Vipr2* and *Adcyap1r1*), by urocortin and corticotropin releasing hormone (acting on the gene product of *Crhr2*), by parathyroid hormone (acting on the receptor encoded by *Pth1r*), or by calcitonin gene-related peptide or adrenomedullin (acting on the gene product of *Calcrl*)[40–44]; all these receptors are $G_s$ coupled[45]. Vasoconstriction, in contrast, is induced by peptide mediators such as endothelin-1, angiotensin II, vasopressin or neuropeptide Y acting on the $G_{i/o}$-, $G_{q/11}$- or $G_{12/13}$-coupled receptors encoded by *Ednra*, *Ednrb*, *Agtr1a*, *Avpra1a* and *Npy1r*[46–48]. The second-largest group of GPCRs with preferential expression in SMsk/SMmes are the orphan receptors, for example, *Gpr19*, *Gpr21*, *Mrgprf*, *Gprc5b*, *Gprc5c*, *Gpr124*, *Gpr126* or *Lphn1*. This suggests that also orphan GPCRs expressed in SMsk are potential targets for modulation of blood pressure, but due to the lack of known agonists/antagonists so far no data are available with respect to their function in regulation of vascular tone or other SMC functions.

Also EC show, depending on their anatomical location, remarkable differences in the GPCR repertoire, and also their responses to acute inflammatory activation differ. Brain EC, for example, were characterized by an upregulation of chemokine receptor *Darc* and orphan receptor *Gpr153*, a response that was absent in lung EC. In line with this finding it was recently shown that DARC is upregulated in brain EC during neuro-inflammation, where it shuttles inflammatory chemokines across the blood-brain barrier[49,50]. The role of GPR153 in activated brain EC, in contrast, is completely unclear. Interestingly, we also detected a small subgroup of brain EC that were characterized by a myeloid-like GPCR expression pattern, but were devoid of any myeloid lineage markers. The function of this subpopulation is currently unclear, but the fact that expression of these GPCRs was negatively correlated with expression of *Icam1* or chemokine receptors involved in leukocyte trafficking, such as *Darc* or *Cxcr7*(ref. 50), suggests that this population is less supportive of leukocyte extravasation than other LPS-activated ECbr. It is also interesting to note that chronic inflammatory activation during atherosclerosis results in other changes than acute activation by LPS, for example, with respect to atypical chemokine receptor *Cxcr7*, or orphan GPCRs *Gpr111*, *Gpr107*, *Gprc5a* and *Gprc5b*. It is also noteworthy that ECao from atherosclerotic aortae showed reduced expression of endothelial genes such as *Cdh5*, *Pecam1*, *Gpr116* or *Gpr56*, suggesting that EC, very much like SMC, undergo dedifferentiation in atherosclerotic aortae *in vivo*.

Taken together, our results show that expression profiling on the single-cell level allows the identification of receptors that are predominantly or selectively expressed on pathologically relevant vascular subpopulations. Understanding the GPCR profile of these cells will not only significantly enhance our knowledge about the pathomechanisms of disease, it will also allow a more selective pharmacological targeting of these cells.

## Methods

**Experimental mice.** All animal experiments were conducted in accordance with the corresponding institutional guidelines and permission by the state of Hessen. C57BL/6J mice, ApoE-deficient mice[25] and the double fluorescent reporter line *Gt(ROSA)26Sor^{tm4(ACTB-tdTomato,-EGFP)Luo}*/J[pos] (dTom/EGFP-reporter) (Stock #007576)[51] were purchased from the Jackson Laboratory. Mouse lines expressing recombinase Cre or the tamoxifen-inducible CreERT2 under control of the EC-specific *Cdh5* promoter (B6.FVB-Tg(Cdh5-cre)7Mlia/J) (*Cdh5*-Cre) or the SMC-specific *Myh11* promoter (B6.FVB-Tg(Myh11-cre/ERT2)1Soff/J) (*Myh11*-CreERT2) were described previously[52,53]. Gprc5b^{tm1a(EUCOMM)Wtsi} (EPD0534_1_A10) mice were obtained from EUCOMM/The Sanger Institute. Gabbr1^{tm1a(KOMP)Wtsi} mice were generated from targeted ES cells (CSD76009, KOMP Repository) by injection into C57BL/6J blastocysts. Mrgprf^{tm1a} mice were generated by targeting v6.5 ES cells using vector Mrgprf^{tm107656a(L1_L2_Bact_P)} (KOMP Repository) and consecutive injection of targeted ES cells into C57BL/6J blastocysts. All lines were kept on the C57BL/6J background and if not otherwise indicated both male and female mice were used at an age of 8–18 weeks. Mice were housed under a 12 h light–dark cycle with free access to food and water and under pathogen-free conditions.

Cre-mediated recombination was induced in *Myh11*-CreERT2 mice by i.p. injection of tamoxifen dissolved in Miglyol 812 (1 mg per mouse per day, Sigma T5648) on 5 consecutive days. To induce atherosclerosis development, ApoE-deficient mice were fed a high-fat diet (21% butter fat, 1.5% cholesterol (Ssniff Cat. TD88137)) for 16 weeks. Animals were killed by i.p. injection of ketamine (180 mg kg$^{-1}$, Pfizer) and xylazine (16 mg kg$^{-1}$, Bayer) and perfused via the left ventricle with phosphate-buffered saline (PBS). In some cases LPS serotype 0111:B4 (Sigma-Aldrich L2630) was injected i.p. at 10 mg kg$^{-1}$ 12 h before killing.

**Single-cell quantitative real-time PCR.** For single-cell expression analysis, ~1,500 cells were loaded in a volume of 5 μl onto the microfluidic C1 Single-Cell Auto Prep System (5–10 μm mRNA Arrays, Fluidigm, San Francisco, CA, USA), followed by RNA isolation and cDNA synthesis according to the manufacturer's protocol. cDNAs derived from visually empty chambers or chambers containing more than one cell were excluded from further analysis; also cDNAs that showed poor expression of reference gene *Hprt* in conventional RT-PCR were excluded. High-throughput quantitative PCR on single-cell cDNAs was performed on 96.96 Dynamic Array IFCs on the BioMark system (Fluidigm) using Sso-Fast EvaGreen Supermix low ROX (BioRad, Hercules, CA, USA) and Delta Gene primer assays (Fluidigm) as listed in Supplementary Table 1. Primers used for the single-cell analysis of human genes are shown in Supplementary Table 2. Only single-cell cDNAs negative for lineage markers *Ptprc* (CD45R, all leukocytes), *Itgam* (CD11b, monocytes/macrophages), *Cd4* (CD4 T cells), *Cd8* (CD8 T cells), *Cd19* (B cells), *Ly6g* (neutrophils), Cdh1 (E-Cadherin, epithelial cells), *Tnni2* (skeletal muscle), but positive for *Gapdh*, *Myh11* (for SMC), or *Cdh5* (for EC) were included into analyses (exception: *Cdh5*[pos]; *Myh11*[low] ECao). Because C1 cDNA samples are potentially contaminated with genomic DNA, intron-spanning primer design was used for all genes. The limit of detection for the BioMark HD System has been estimated to be at a Ct value of 24 cycles (Limit of detection (LoD) Ct); all sample Ct values were therefore subtracted from the (LoD) Ct using the formula: gene expression $= 2^{(LoD\ Ct - sample\ Ct)}$[54].

**Tissue digestion and cell sorting.** Animals were killed, perfused with PBS and different tissues (lung, skeletal muscle of lower limbs, brain, mesentery, whole aorta including common carotid arteries) were dissected, minced and enzymatically digested for 90–120 min while shaking at 37 °C in a digestion mix containing collagenase II (2 mg ml$^{-1}$; Worthington), elastase-I (0.04 mg ml$^{-1}$; Sigma), and DNase I (5 U ml$^{-1}$; New England Biolabs). For lung digestion, also dispase II (1.2 units per ml; Sigma) was added to the mix. Cell suspensions were serially filtered through 70 and 40 μm cell strainers followed by washing twice with PBS. Cells from Cdh5-Cre[pos]; dTom/EGFP-reporter[pos] mice or Myh11-CreERT2[pos]; dTom/EGFP-reporter[pos] mice were sorted on a JSAN cell sorter (Bay Biosciences, Japan) based on their EGFP expression; in other cases antibodies directed against CD31 (Serotec MCA2388PE) or CD144-PE (ebioscience #12144180) were used to identify EC.

**Bioinformatic analysis of single-cell RT-PCR data.** RT-PCR-based expression values for individual cells were analysed by customized functions from the RaceID R package[18]. For k-means clustering, we utilized the clustexp and clustheatmap functions; cluster specific genes ($P < 0.05$, according to expected transcript count probability from binomial testing, see Methods section from[18] were exported by a customized clustdiffgenes function. Graphs show fold changes ($>1.5$ or $<0.7$) from cluster specific genes. T-SNE plots were generated by the plottsne function. Heat maps based on RT-PCR expression data were generated in Perseus software[55]. The correlation between expression of individual genes was determined using Spearman's rank correlation coefficient; graphical representations of Spearman's rank correlation coefficients were generated using Cytoscape[56]. The width of a connector indicates the strength of the correlation; only correlation coefficients $>0.3$ are displayed.

To investigate the potential role of transcription factors (TFs) in cell type-specific GPCR sets, we utilized the pscan tool[57]. We investigated the promoter region of GPCRs within a given set at a size of $-950$ to $+50$ bp around the respective transcription start site and tested for significantly ($P < 0.05$) overrepresented/underrepresented TF binding sites listed in the current Jaspar 2016 release (http://jaspar.genereg.net/)[58] compared to a global promoter set of the mouse organism. Resulting $P$ values were z transformed and illustrated in a multi column heat map generated by the heatmap2 R function (package gplots, version 3.01).

**Single-cell mRNA sequencing and expression analysis.** Single-cell transcriptome analysis was performed on a C1 Autoprep station (Fluidigm) using 5–10 μm mRNA arrays and standard protocols. In total, 15,000 events sorted on a JSAN swift sorter were suspended in 10 μl resuspension buffer (PBS + 0.5% bovine serum albumin + 2 nM EDTA, sterile filtered) and further diluted (6:4) in suspension buffer. A volume of 3 μl was loaded to the C1 Array while remaining cells ($\sim$10,000) were used for total RNA isolation using μRNeasy micro kit (Qiagen) combined with on-column DNase digest (Qiagen). RNA was quantified by Qubit HS RNA Assay (Thermo Fisher) and used for tube control. Pre-amplification and cDNA synthesis were done with Smarter ultra-low RNA Kit (Clontech) without changes to the standard protocol. cDNA obtained from C1 system was quantified using PicoGreen assay (Thermo Fisher) and diluted to 0.1–0.3 ng μl$^{-1}$. Picking of positive cDNA samples and library preparation were done with NGS Express pipetting robot (Perkin Elmer) using custom made protocols. Barcoded and amplified libraries were pooled in groups of 16 libraries for final SRRI-bead based cleanup and quantified by Qubit HS dsDNA Assay (Thermo Fisher) and BioAnalyzer 2100 HS DNA Assay (Agilent). Sequencing was performed on Nextseq500 Sequencer (Illumina) using v2 chemistry and High Output Flow Cell with 75 bp single end protocol. Raw data files were trimmed using Reaper with a quality cutoff of 53 (length 20 bp, prefix 50 bp) and a minimum clean read length of 50 bp. Reads were then mapped to the genome (mm10) using STAR. FeatureCounts was used to assign reads to genes from the Gencode vM6 annotation. Genes with less than 50 counts across all cells were discarded before normalization. The knn.error.models function from SCDE[59] was used to construct cell-specific error models by evaluating the gene counts for a observed cell versus the expected gene counts from the k most similar cells. The parameter k was chosen to resemble roughly three subpopulations. Estimated fragments per million mapped reads (FPM, in log scale) were then obtained using the scde.expression.magnitude function and are given in Supplementary Data 1.

**NanoString analysis of GPCR expression in bulk RNA.** NanoString analyses were performed as described previously[60,61]. In brief, 250–500 ng RNA from sorted cells was applied in a total volume of 30 μl in the assay. Barcodes were counted for $\sim$1,150 fields of view per sample. Counts were first normalized to the geometric mean of the positive control spike count, then a background correction was done by subtracting the mean + two s.d. of the eight negative control counts for each lane. Data were not normalized to reference genes because none of the reference genes showed sufficiently stable expression in all cell types according to the geNorm algorithm. Values that were <20 were fixed to background level.

**Cell culture and transfection.** Isolated human aortal SMC (passage 1) were obtained from Cell Applications (Cat. No. 354-05a; San Diego, CA, USA) and Innoprot (Cat. No. P10456; Derio, Spain). Cells were thawed and immediately processed for expression analysis without further culture. Primary murine aortic SMC were isolated as described above and cultured in collagen I-coated (100 μg ml$^{-1}$; Corning) 24-well plates for 7 days. Cells were transfected on day 0 with siRNAs directed against murine Gprc5b (Mm_Gprc5b_6; target sequence: 5′-TCGGGCCTACATGGAGAACAA-3′) or AllStars negative control siRNA using Lipofectamine RNAiMAX. All siRNAs were obtained from Qiagen.

**RNA isolation and RT-PCR for cell pool analysis.** RNA for cell pool NanoString analysis was isolated with the RNeasy Micro or Mini Kit (Qiagen) according to the manufacturer's instructions. Gene expression in cultured SMao was analysed using the Universal ProbeLibrary System (Roche, Basal, Switzerland) with the following primer pairs:
Actb (probe #15): Fwd: 5′ AAATCGTGCGTGACATCAAA 3′/Rev: 5′ TCTCCAGGGAGGAAGAGGAT 3′;
Acta2 (probe #80): Fwd: 5′ TCTGGACTTTGAAAATGAGATGG 3′/Rev: 5′ CCCGTCAGGCAGTTCGTA 3′
Icam1 (probe #20): Fwd: 5′ CGTGGGGAGGAGATACTGAG 3′/Rev: 5′ GTGATCTCCTTGGGGGTCCTT 3′
Vcam1 (probe #50): Fwd: 5′ CCGGTCACGGTCAAGTGT 3′/Rev: 5′ CAGATCAATCTCCAGCCTGTAA 3′
Gprc5b (probe #64): Fwd: 5′ GCCTTCTCAATGGATGAACATA 3′/Rev: 5′ CAAGCTGCTGGGCTTCTT 3′
Expression of Myh11, Mki67 and Il6 was determined using LightCycler 480 SYBR Green I Master (Roche) and primers listed in Supplementary Table 1.

**Immunohistochemical analyses.** Aortic arches were cryosectioned transversally (10 μm) and fixed with ice-cold acetone for 10 min. O.C.T. tissue freezing medium (Sakura) was removed by washing with PBS and sections were immunostained with biotinylated anti-α-smooth muscle actin (αSMA) antibody (1:100, Abcam ab125057) and chicken anti-ß-galactosidase (1:400, Abcam ab9361) overnight. After washing, antibody binding was detected using strepatividin-alexa 647 (ThermoFischer, S32357) and goat anti-chicken IgG (H + L)-Alexa-568 (ThermoFischer, A11041) (both 1:200). 4′,6-diamidin-2-phenylindol (DAPI) was used to label cell nuclei.

**Flow cytometric analysis.** β-galactosidase expression in murine aortic SMC was analysed using the FlouReporter lacZ flow cytometry kit (ThermoFischer) with propidium iodide to identify viable cells and anti-αSMA antibody (ebioscience, #53976080) to identify SMC. For antibody-mediated detection of GPCR expression, aortae of wild-type mice were digested, fixed in 0.01% formaldehyde and permeabilized using Tween20 (0.5% v/v in PBS), followed by incubation of resulting single-cell suspensions with PE- or FITC-labelled antibodies directed against αSMA (ebiosciences) and APC-labelled antibodies directed against receptors Cmklr1 (Miltenyi), Ccrl2 (R&D), Cxcr7 (R&D), Celsr2 (R&D) (in all cases 10 μl antibody per 10$^6$ cells per 200 μl). For each GPCR-specific antibody the corresponding isotype control was used. Analyses were performed on a FACS Canto II (Becton Dickinson) and data were analysed by FlowJo software (Tree Star).

**General statistical analyses.** Data are presented as means ± s.e.m. if not otherwise indicated. Comparisons between two groups were performed using two-sample $t$-test; normalized data (control group set to 1) were analysed by one-sample $t$-test. $P$ values are indicated as follows: *$P < 0.05$; **$P < 0.01$; ***$P < 0.001$.

**Data availability.** Single-cell RNA-Seq data have been deposited in GEO database (https://www.ncbi.nlm.nih.gov/geo) under the accession code GSE97955. Full data sets generated during the current study are available from the corresponding author on reasonable request.

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

## Acknowledgements

We thank Mona Khan and Peter Mombaerts for their generous help with expression analyses. We furthermore thank Karin Jäcklein, Ulrike Schlapp and Claudia Ullmann for expert technical assistance.

## Author contributions

H.K. performed most of the experiments, analysed and discussed data, and wrote parts of the manuscript, J.C. performed expression analyses and in vitro experiments, P.S., D.T., S.G. and S.C. helped with expression analyses, M.L., J.P. performed bioinformatic analyses, R.C. generated Mrgprf reporter mice, J.A.-J. helped with immunostaining, S.O. initiated the study, discussed data and commented on the manuscript, N.W. initiated and supervised the study, analysed and discussed data and wrote the manuscript.

## Additional information

Competing interests: The authors declare no competing financial interests.

