## [Peer Review File · Nature Communications]

Reviewers' comments:

Reviewer #1 (expert in GPCRs and vascular cells)

Remarks to the Author:

This was a study using microfluidic-based single-cell GPCR expression analysis in freshly isolated healthy and diseased VSMCs and ECs. Single cell RT-PCR and RNA sequencing was used to characterize selected GPCR expression in endothelial and smooth muscle cells. The authors found that RT-PCR was superior to RNA sequencing in this population, there was significant heterogeneity in the expression of GPCRs and they were also able to identify distinct cell populations, such as dedifferentiated SMCs. They observed changes in GPCR expression in the setting of sepsis and atherosclerosis. Earlier studies have studied the GPCR single expression primarily in the brain (Spaethling et al., FASEB J. 2014; Manteniotis et al., PLoS One 2013; Hanchate et al., Science 2015). I think the results of this study would be of interest to a wide audience. These findings, though, are primarily descriptive and do not provide significant insights into disease processes and it is unclear how heterogeneity contributes to the functions of these vascular cells (except for the case of the dedifferentiated smooth muscle cells where dedifferentiation were associated with binding of specific transcription factors). The limitations in mechanistic insight limit enthusiasm for the manuscript.

Reviewer #4 (expert in single cell transcriptomics)

Remarks to the Author:

The manuscript "Single-cell expression profiling reveals heterogeneity and functional patterning of GPCR expression in the vascular system" by Kaur et al analyzed the expression patterns of G-protein-coupled receptor (GPCR) in single vascular smooth muscle cells (SMC) and endothelial cells (EC) isolated from various sites on healthy and disease conditions by using a microfluidic-based single cell GPCR expression analysis. They found that GPCR expression is highly heterogeneous in all analyzed cell types. This was shown by both the single cell qPCR and single cell RNA-seq techniques, while the authors found that their single cell qPCR assay showed higher frequencies of expression than the single cell mRNA-seq assay. The authors found that different types of SMCs or ECs have distinct GPCR expression patterns. Further, the authors identified changes of GPCR expression patterns after inflammatory activation and they also identified and verified a dedifferentiating SMC subpopulations in healthy SMCs. In general, this study provides novel and valuable insights into the complex expression of GPCRs in EC and SMC at single cell resolution.

Major:

1. The authors mainly used a microfluidic-based single cell GPCR expression assay for analysis. They have also performed the single cell RNA-seq method and found their single qPCR assay showed higher frequencies of GPCR expression than the single cell mRNA-seq assay for nearly all GPCRs. The authors explained that the better performance of the single-cell RT-PCR may be due to its use of target-specific pre-amplification of mRNAs, while pre-amplification for mRNA sequencing is unbiased. However, the authors should exclude the possibility that the RT-PCR assay amplifies off-target sites, particularly that the sequences of different GPCRs genes may be highly similar. This is less possible to happen in the single RNA-seq method with its base-resolution character. The authors have verified the results of the RT-PCR assay by showing that the single cell qPCR results were consistent with the results of the GPCR reporter mice. However, they should directly address the off-target issue by sequencing (high throughput sequencing) the PCR product of the single-cell RT-PCR assay.
2. The authors found that different types of SMCs or ECs have distinct GPCR expression patterns and can be grouped into distinct clusters by clustering analysis. The authors should give a list to clearly show how many experiments and how many cells in each experiment have been analyzed for each

SMC and EC cell types. In Figure 3G and 4C, for a certain cluster, have the cells been analyzed by more than one independent experiments to exclude the batch effect?

3. The authors also found that GPCR expression was highly heterogeneous within each cell type. However, in most conditions, it seems that the authors have not identified functional cell subpopulations. I am not convinced that the intra-cell type heterogeneous GPCR expression has important biological functions in general. The authors should be careful to claim this. It can be helpful to verify the heterogeneous GPCR expression at the protein level if suitable antibodies are available. If you can detect the heterogeneously expressed GPCR gene at the protein level, it is more convincing that the gene could be functional.

4. The authors have identified a small subpopulation of dedifferentiating SMC from healthy SMao (Figure 8). They have also performed the single cell RNA-seq method for SMao. Could this small subpopulation also be identified from the single cell RNA-seq data and were the results between the single cell RT-PCR and single cell RNA-seq consistent?

Reviewer #4 (expert in single cell transcriptomics)

Remarks to the Author:

Major:

1. The authors mainly used a microfluidic-based single cell GPCR expression assay for analysis. They have also performed the single cell RNA-seq method and found their single qPCR assay showed higher frequencies of GPCR expression than the single cell mRNA-seq assay for nearly all GPCRs. The authors explained that the better performance of the single-cell RT-PCR may be due to its use of target-specific pre-amplification of mRNAs, while pre-amplification for mRNA sequencing is unbiased. However, the authors should exclude the possibility that the RT-PCR assay amplifies off-target sites, particularly that the sequences of different GPCRs genes may be highly similar. This is less possible to happen in the single RNA-seq method with its base-resolution character. The authors have verified the results of the RT-PCR assay by showing that the single cell qPCR results were consistent with the results of the GPCR reporter mice. However, they should directly address the off-target issue by sequencing (high throughput sequencing) the PCR product of the single-cell RT-PCR assay.

This is of course a valid point, and we sequenced the single-cell RT-PCR products of all GPCRs expressed in aortic smooth muscle cells at a frequency of more than 10% (as shown in Figures 2A and Suppl. Fig. 2 of the manuscript). Each amplicon was sequenced in forward and reverse direction, and the obtained sequences were aligned to the predicted amplicon sequence. Figure 1 of this letter shows an example of sequencing results and the corresponding alignment; the alignments for all sequences are shown in Table 1 of this letter.

Figure 1: Exemplary analysis of results obtained by sequencing of single-cell amplicons (here for the alpha-1A adrenergic receptor Adra1a): Single-cell PCR products were sequenced with the same forward and reverse primers that were used for amplification, the respective chromatograms as well as the alignment of sequences to the predicted Adra1a amplicon are shown above (forward primer, matching base pairs in red) and below (reverse primer, matching base pairs in green) the aligned sequences. The sequence obtained from the reverse primer is for convenience shown reverse and complementary.

Table 1: Alignment of sequencing results

Adrala_F	-----ctgctggctgccattcttctcctcgtgatgccattgggtccttc-----
Amplicon	gagaagaaagccgccaagacgctgggcatttggtgggatgcttcgtcctctgctggctgccattcttctcctcgtgatgccattgggtccttccttcccgaat
Adrala_R	-----gccaaagacgctgggcatttggtgggatgcttcgtcctc-----
Adralb_F	-----ctccaacctaaagccccggacgccgtattcaaggtagtgttctggctgggc--
Amplicon	atggttgctcccttcttcatcgcctctcccacttggtccctggtctccaacctaaagccccggacgccgtattcaaggtagtgttctggctgggcta
Adralb_R	atggttgctccct-----
Adrald_F	-----ggttctctgttcctcagctgaaacctcag-----
Amplicon	ttcgtcctgtgctggttccctttttcttcgtcctgcctctgggttctctgttcctcagctgaaacctcagagggcgctc
Adrald_R	ttcgtcctgtgctggttccctttttct-----
Agtr1a_F	-----gtgtctgagaccaactcaaccaga
Amplicon	taactcacagcaaccctccaagaaagccatcacccagatcaagtgcatthtgaacagtgtctgagaccaactcaaccaga
Agtrala_R	taactcacagcaaccctccaagaaagccatca-----
Bdkrb2_F	-----tgcccaccgggctcctttggcatcgaatggttcaacgtcaccac-----
Amplicon	aatgcctgctcctggaagctactcgggtttctgtcggatgagcccatgcccaccgggctcctttggcatcgaatggttcaacgtcaccacacaag
Bdkrb2_R	--atgcctgctcctggaagctactcgggtttctgtcggatgagcccat-----
Calcr1_F	-----ctgggacggatggctatgctggaatgacgttg
Amplicon	atgcaggacccattcaacaagcagaaggcctttactgcaataggacctgggacggatggctatgctggaatgacgttg
Calcr1_R	atgcaggacccattcaacaagcaga-----
Ccr12_F	-----agcctccgatggataactacacagtggccc
Amplicon	caagcaacctgctcaaacgacgctgttttgtccggatgagcaaggacagcctccgatggataactacacagtggccc
Ccr12_R	caagcaacctgctcaaacgacgc-----
Cd97_F	-----ggacc-aaaggctggaattgatcaccaaggtggg
Amplicon	tgaccagctttgccatcctaattggctcagtaccatgtgcaaggacaaaggctggaattgatcaccaaggtggg
Cd97_R	tgaccagctttgccatcctaattggctcagtacc-----
Celsr2_F	-----tggtgacaactgtacaaatgtgtgtgacctgaacca
Amplicon	agcaatgactgggacagctattctttagctgtgttctaggtactatggtgacaactgtacaaatgtgtgtgacctgaacca
Celsr2_R	agcaatgactgggacagctattctttag-----

Table 1 (cont.): Alignment of sequencing results

Cmklr1_F	-----ctcaaagagatggagtagcagcgttacaacgac
Amplicon	gtaacagaccagccaaggacca-ggactggagtctgttctacaacggtgaacagtgaaaggtctcaaagagatggagtagcagcgttacaacgac
Cmklr1_R	gtaacagaccagccnaggaccaaggacggag-----
Crhr2_F	-----tttcaggtccctacacctactgcaacacgaccttgg
Amplicon	gtggacacttttggagcagtactgccacaggaccacaactgggaatttttcaggtccctacacctactgcaacacgaccttgg
Crhr2_R	gtggacacttttggagcagtactgccacaggac-----
Cxcr7_F	-----caaaccacagcccaggaagccctgaggtcacttggtcgctctcctcaagac
Amplicon	acaaactgctcagcactgaaggagcctgcagcgctcaccgtcaggaagcaaaccacagcccaggaagccctgaggtcacttggtcgctctcctcaagac
Cxcr7_R	acaaactgctcagcactgaaggagcctgcagcgctcaccgtcaggaag-----
Ednra_F	-----gatgtgaaggactggtggctctt
Amplicon	cataggacctgcatgctcaacgccacgtccaagttcatggagttttaccaagatgtgaaggactggtggctctt
Ednra_R	cataggacctgcatgctcaacgccacgtccaagt-----
F2r_F	-----agaggacagatgctacggtga
Amplicon	gcggtcccttgtgtcttcccggctccctatgagccagccagaatcagagaggacagatgctacggtga
F2r_R	gcggtcccttgtgtcttcccgc-----
F2r11_F	-----ggaccgagaaccttgcaccgggacgcaacaacagtaaaggaagaagct-----
Amplicon	ggctgctgggaggtatcaccttctggcggcctcggtctcctgcagcggaccgagaaccttgcaccgggacgcaacaacagtaaaggaagaagcttattggcagat
F2r11_R	-----ctcggtctcctgcagccggaccgagaaccttgcaccgggacgcaacaaca-----
Gabbr1_F	-----tcgtgggacttttctatgagaccgaagccc
Amplicon	agatccagctgtgcctgttaaaaaacctgaagcgtcaagatgctcgaatcatcgtgggacttttctatgagaccgaagccc
Gabbr1_R	agagccagctgtgcctgntaaaaacctgaagcgtc-----
Gpr107_F	-----cgcagggctttccgattgaaggctgggctgttga
Amplicon	gtctctctcgttgggtgttccatgcaatcgactaccactacatctcctcgcagggctttccgattgaaggctgggctgttga
Gpr107_R	gtctctctcgttgggtgttcca-gcaa-----
Gpr108_F	-----tcaactgtgatgatccgggagaagaatcca
Amplicon	tccacaactgtcacaactccatcccaggccaggagcagccattcgacctcaactgtgatgatccgggagaagaatcca
Gpr108_R	ccacaactgtcacaac-----

Table 1 (cont.): Alignment of sequencing results

```
Gpr124_F -----ttccttgcgtcaactgctctggat
Amplicon tCACgctcaccAActacCAaatggtttgtcaagcggtgggcatcactctgcactacttCtCcttgcgtcaactgctctggat
Gpr124_R tCACgctcaccAActacCAaatggtt-gtca-----

Gpr125_F -----tattctacccttgccacggt
Amplicon tgggaggaataaccagaccagaaatgccagcgctctgtcaagcagttgggatcattcttcaattattctacccttgccacggt
Gpr125_R tgggaggaataaccagaccagaaatgccag-----

Gpr133_F -----agaacggtgggt-----
Amplicon tccgcttgcccaataaatccctctcagaggaaacggcgctgaacctcacagagaccttcttaagaacggtgggtgaggt
Gpr133_R tccgcttgcccaataaatccctctcagag-----

Gpr137_F -----ctctggcagttggtatggtgccatcgga
Amplicon gctcctgggagcatagccggagtgagagcacacagcatgtccggcagcctgggctctggcagttggtatggtgccatcgga
Gpr137_R gctcctgggagcatagccggagtgagagc-----

Gpr137b_F -----cagtggtcaggtaactgccattggtgtcacogtcatctt-
Amplicon tgtccctggccaacatctacttgagggtcaaaagggtcatcagtggtcaggtaactgccattggtgtcacogtcatcttg
Gpr137b_R -gtccctggccaacatctac-gtggaggtcaaa-----

Gpr153_F -----ccccgacatggtattggagcgctctctt
Amplicon gccaacgacgaagattctgacaatgagaccagtcctagagggcagcatctccccgacatggtattggagcgctctctt
Gpr153_R -ccaacgacgaagattctgacaatgagaccagtc-----

Gpr21_F -----atgcagaatcacagctggtcagaggagactcattacaactcctgc-
Amplicon gcattgcaaggctttcggttaaggatgtattgtggcttttgtttggatttcagcatgcagaatcacagctggtcagaggagactcattacaactcctgct
Gpr21_R -----gcaaggctttcggttaaggatgtattgtggcttttgtttggatttcagca-----

Gpr4_F -----tagcagtcg-cagctctcaatgcagtg-gtgagtggaacaacttcatctccact
Amplicon cgggaccaagtCagagacgcccggctgcccagcccagccgaggagcaggcctagcaatctcagctctcaatgcagtcgtgagtggaaCaacttcatctccactcctcag
Gpr4_R -----agccgaggagcaggcctagcagtcg-cagctctcaatgcagtg-gtgagtgaa-----

Gpr64_F -----tgtgtattctgggacttgggcagaa
Amplicon gtgacagtcgcaactgaaacacatcaaccCaagtccggatgacttaactgtgaaatgtgtattctgggacttgggcagaa
Gpr64_R gtgacagtcgcaactgaaacacatcaacc-----
```

Table 1 (cont.): Alignment of sequencing results

Gprc5b_F	-----ccgatcagcagtgggcttttccaatggaagcttggagcaaagatc-
Amplicon	cgggcctacatggagaacaaggccttctcaatggatgaacataacgcagctctccgatcagcagtgggcttttccaatggaagcttggagcaaagatct
Gprc5b_R	cgggcctacatggagaacaaggccttctcaatggatgaacataacgca-----
Htr2a_F	-----ggtaccggtggcctttgccagcaagctctgt
Amplicon	gctgctgggtttccttgtcatgcccggtccatggttaaccatcctgtatgggtaccggtggcctttgccagcaagctctgt
Htr2a_R	gctgctgggtttccttgtcatgcccggtccatgt-----
Lgr4_F	-----gtaattctatcttctgttatcccggatggagca-
Amplicon	ctcaggctattaaagcccttcccagccttaaagagctgggatttcacagtaattctatcttctgttatcccggatggagcat
Lgr4_R	ctcaggctattaaagcccttcccagccttaa-----
Lgr6_F	-----gctgcatctacataacaaccgcatccagcatg
Amplicon	tatccgccacatccctgactatgccttccagaacctcaccagtcttgtgggtgctgcatctacataacaaccgcatccagcatg
Lgr6_R	tatccgccacatccctgactatgccttccagaacc-----
Lpar1_F	-----gactgtggtcattgtgcttggtgcctttattgtctgctggactccgg
Amplicon	agttctggaccaggaggaatcgggacaccatgatgagccttcgaaagactgtggtcattgtgcttggtgcctttattgtctgctggactccgg
Lpar1_R	agttctggaccaggaggaatcgggacaccatgatgagccttc-----
Lpar4_F	-----ttctctcatctagcacactctttcttgggcaactcaattgaggaaac
Amplicon	agtgcgagttgccagtttacacgtttattagctaactatctacaggcatgagcacattctctcatctagcacactctttcttgggcaactcaattgaggaaactctctga
Lpar4_R	-----acgtttattagctaactatctacaggcatgagcacattctctcatctag-----
Lphn1_F	-----ggtgaaagttgtcttattctctacaacaacctgggcctcat
Amplicon	tctccgccaacacc-atcaagcagaacagccgcaacggtgtggtgaaagttgtcttattctctacaacaacctgggcctctt
Lphn1_R	tctccgccaacaccgatcaagcagaacagccgcaa-----
Lphn2_F	-----gacaacctctgagagctgaggccttggaaatcctgga
Amplicon	gaagacatgcagggttaccttaaggcaattgtggacacggtagacaacctctgagagctgaggccttggaaatcctgga
Lphn2_R	gaagacatgcagggttaccttaaggcaat-----
Mrgprf_F	-----tgtcctggtatgagcaggctcgga
Amplicon	agatggccggaactgttcatgggaagctcactccaccaaccagaacaagatgtgtcctggtatgagcaggctcgga
Mrgprf_R	aga_ggcccggaactgttcatgggaagctcagcgc-----

Table 1 (cont.): Alignment of sequencing results

Npy1r_F	-----aggagaaacaacatgatggacaagatccggg
Amplicon	ttcggcccactctgctttatattcatatgctacttcaagatatacattcgccttgaaaaggagaaacaacatgatggacaagatccggg
Npy1r_R	ttcggcccactctgctttatattcatatgctacttcaa-----
P2ry2_F	-----gacctggaaccctggaatagcacc
Amplicon	gagcatcctcaccacctcaagagcaggagctgatcaggtccagggcaatggcagcagacctggaaccctggaatagcacc
P2ry2_R	gagcatcctcaccacctcaagagcaggagctgatcag-----
P2ry6_F	-----cgagcataggaaaggctgacaggcag
Amplicon	ccaaatctggcacttcctcctaaaacatcttccatcttgcatgagacagactctccgagcataggaaaggctgacaggcag
P2ry6_R	ccaaatctggcacttcctcctaaaacatcttcc-----
Ptger3_F	-----ctattgataatgatggtgaaaatgatcttcaatcagatgtcggttgagc
Amplicon	cagctcatggggatcatgtgtgtgctgtccgtctgttggtcgccgctattgataatgatggtgaaaatgatcttcaatcagatgtcggttgagc
Ptger3_R	cagctcatggggatcatgtgtgtgctgtccgtctgttggtcgccgct-----
Ptgir_F	-----ccgaggcttcaactcaggccatcgcc
Amplicon	attctgctggccctcatgaccgcatcatggcgtgtgctccctgacctcatgatccgaggcttcaactcaggccatcgcc
Ptgir_R	attctgctggccctcatgaccgcatcatggcgtgtgc-----
Pth1r_F	-----ctcaacttcatcctctttatcaacatcatccgggtgct
Amplicon	tgggcacaagaagtggatcatccaggtgcccatcctggcatctggttgctcaacttcatcctctttatcaacatcatccgggtgct
Pth1r_R	tgggcacaagaagtggatcatccaggtgctcngatcc-----
Tbxa2r_F	-----ttcatcatgcagactttggtgcagacaccacct-
Amplicon	ttcagctcgtgggcatcatggtggtggccacggtgtgtggatgccttgctggcttcatcatgcagactttggtgcagacaccacctg
Tbx2ar_R	ttcagctcgtgggcatcatggtggtggccacggtgtgtggatgc-----

Table 1: Alignment of sequences obtained by sequencing of single-cell amplicons with forward (F) and reverse (R) primers. Non-matching base pairs are highlighted in yellow.

Table 1 shows that all sequences obtained from forward or reverse primers matched the predicted amplicon sequences, confirming that the correct mRNA was amplified. To furthermore exclude that mRNAs of highly similar GPCRs were amplified, we compared the sequencing results to the murine refseq_rna database (version mm10) using the Nucleotide BLAST tool (<https://blast.ncbi.nlm.nih.gov/Blast.cgi>). Table 2 shows for each of the sequencings the three best blast hits; the E-value indicates the likelihood that the respective match occurred by chance. For an extended list of blast results (limited to 20 hits per query), including start/end point of homology and hit sequence, please see the attached file “Supplemental Table 1 for Reviewers”.

Table 2: Blast analysis of single-cell transcript sequencing results.

Gene	Que-ry	Hit	E-value	% match	length align-ment
Adra1a	For	Mm_adrenergic_receptor_alpha_1a_(Adra1a),_mRNA	3,90E-17	100	43
		Mm_adrenergic_receptor_beta_1_(Adrb1),_mRNA	0,13	95,24	21
		Mm_leucine_rich_repeat_and_fibronectin_type_III_extracellular_1_(Elfn1),_mRNA	0,13	100	17
	Rev	Mm_adrenergic_receptor_alpha_1a_(Adra1a),_mRNA	8,10E-15	100	39
		Mm_trace_amine-associated_receptor_5_(Taar5),_mRNA	0,007	100	19
		PREDICTED: Mm_RIKEN_cDNA_4631423B10_gene_(4631423B10Rik),_miscRNA	0,43	100	16
Adra1b	For	Mm_adrenergic_receptor_alpha_1b_(Adra1b),_mRNA	2,21E-22	100	52
		Mm_RIKEN_cDNA_B430010I23_gene_(B430010I23Rik),_non-coding_RNA	0,67	100	16
		Mm_cAMP_responsive_element_binding_protein-like_2_(Crebl2),_mRNA	2,7	91,3	23
	Rev	Mm_adrenergic_receptor_alpha_1b_(Adra1b),_mRNA	0,58	100	14
		Mm_glutamate_receptor_ionotropic_AMPA3_(alpha_3)_(Gria3),_mRNA	2,3	100	13
		Mm_striatin_calmodulin_binding_protein_(Strn),_mRNA	2,3	100	13
Adra1d	For	Mm_adrenergic_receptor_alpha_1d_(Adra1d),_mRNA	3,35E-10	100	31
		Mm_arrestin_beta_2_(Arrb2),_mRNA	0,3	100	16
		Mm_small_nucleolar_RNA_C/D_box_1A_(Snord1a),_small_nucleolar_RNA	1,2	100	15
	Rev	Mm_adrenergic_receptor_alpha_1d_(Adra1d),_mRNA	1,68E-08	100	28
		Mm_adrenergic_receptor_alpha_2c_(Adra2c),_mRNA	0,001	92,86	28
		Mm_endothelin_receptor_type_A_(Ednra),_mRNA	0,24	95	20
Agtr1a	For	Mm_angiotensin_II_receptor_type_1a_(Agtr1a),_mRNA	7,97E-07	100	25
		Mm_NCK_interacting_protein_with_SH3_domain_(Nckipso),_mRNA	2,9	94,44	18
		Mm_UDP-N-acetyl-alpha-D-galactosamine:polypeptide_N-cetylgalactosaminyltransferase	2,9	100	14
	Rev	Mm_angiotensin_II_receptor_type_1a_(Agtr1a),_mRNA	9,01E-11	100	32
		Mm_ubiquitin_specific_peptidase_37_(Usp37),_mRNA	0,02	100	18
		Mm_histone_cluster_3_H2ba_(Hist3h2ba),_mRNA	0,08	100	17
Bdkrb2	For	Mm_bradykinin_receptor_beta_2_(Bdkrb2),_mRNA	7,02E-19	100	46
		PREDICTED: Mm_uncharacterized_LOC100862575_(LOC100862575),_miscRNA	2,2	100	15
		Mm_cytochrome_P450_family_11_subfamily_b_polypeptide_2_(Cyp11b2),_nuclear_g	2,2	100	15
	Rev	Mm_bradykinin_receptor_beta_2_(Bdkrb2),_mRNA	3,27E-21	100	50
		Mm_ankyrin_repeat_domain_29_(Ankr29),_mRNA	0,041	100	18
		PREDICTED: Mm_acetyl-CoA_carboxylase_1-like_(LOC100862524),_mRNA	0,64	100	16
Calcr1	For	Mm_calcitonin_receptor-like_(Calcr1),_mRNA	9,01E-11	100	32
		PREDICTED: Mm_predicted_gene_10847_(Gm10847),_miscRNA	5	100	14
		PREDICTED: Mm_predicted_gene_10847_(Gm10847),_miscRNA	5	100	14
	Rev	Mm_calcitonin_receptor-like_(Calcr1),_mRNA	2,22E-07	100	26
		Mm_coiled-coil_domain_containing_90A_(Ccdc90a),_mRNA	0,21	100	16
		Mm_transmembrane_and_coiled-coil_domains_4_(Tmco4),_mRNA	0,81	100	15
Ccr12	For	Mm_chemokine_(C-C_motif)_receptor-like_2_(Ccr12),_mRNA	1,24E-09	100	30
		PREDICTED: Mm_uncharacterized_LOC100048499_(LOC100048499),_miscRNA	1,1	100	15
		Mm_potassium_channel_tetramerisation_domain_containing_17_(Kctd17),_mRNA	1,1	100	15
	Rev	Mm_chemokine_(C-C_motif)_receptor-like_2_(Ccr12),_mRNA	2,83E-06	100	24
		Mm_T-box18_(Tbx18),_mRNA	2,6	100	14
		Mm_T-box_20_(Tbx20),_transcript_variant_1,_mRNA	2,6	100	14
Cd97	For	Mm_CD97_antigen_(Cd97),_transcript_variant_1,_mRNA	5,56E-09	100	29
		Mm_CD97_antigen_(Cd97),_transcript_variant_3,_mRNA	5,56E-09	100	29
		Mm_CD97_antigen_(Cd97),_transcript_variant_2,_mRNA	5,56E-09	100	29
	Rev	Mm_CD97_antigen_(Cd97),_transcript_variant_1,_mRNA	2,41E-11	100	33

		Mm_CD97_antigen_(Cd97),_transcript_variant_3,_mRNA	2,41E-11	100	33
		Mm_CD97_antigen_(Cd97),_transcript_variant_2,_mRNA	2,41E-11	100	33
Celsr2	For	Mm_cadherin_EGF_LAG_seven-pass_..._(Celsr2),_transcript_variant_1,_mRNA	1,15E-13	100	37
		Mm_cadherin_EGF_LAG_seven-pass_..._(Celsr2),_transcript_variant_2,_mRNA	1,15E-13	100	37
		Mm_MOB_kinase_activator_3C_(Mob3c),_mRNA	0,099	95,24	21
	Rev	Mm_cadherin_EGF_LAG_seven-pass_..._(Celsr2),_transcript_variant_1,_mRNA	4,58E-09	100	29
		Mm_cadherin_EGF_LAG_seven-pass_..._(Celsr2),_transcript_variant_2,_mRNA	4,58E-09	100	29
		Mm_centromere_protein_W_(Cenpw),_mRNA	0,26	100	16
Cmklr1	For	Mm_chemokine-like_receptor_1_(Cmklr1),_mRNA	6,45E-12	100	34
		Mm_expressed_sequence_AI646023_(AI646023),_mRNA	1,4	100	15
		Mm_PHD_finger_protein_16_(Phf16),_mRNA	1,4	100	15
	Rev	Mm_chemokine-like_receptor_1_(Cmklr1),_mRNA	0,005	95,45	22
		Mm_multiple_endocrine_neoplasia_1_(Men1),_transcript_variant_4,_mRNA	4,7	94,12	17
		Mm_multiple_endocrine_neoplasia_1_(Men1),_transcript_variant_2,_mRNA	4,7	94,12	17
Crhr2	For	Mm_corticotropin_releasing_hormone_receptor_2_(Crhr2),_mRNA	1,15E-13	100	37
		PREDICTED:_Mm_RIKEN_cDNA_D130079A08_gene_(D130079A08Rik),_miscRNA	1,6	100	15
		PREDICTED:_Mm_RIKEN_cDNA_D130079A08_gene_(D130079A08Rik),_miscRNA	1,6	100	15
	Rev	Mm_corticotropin_releasing_hormone_receptor_2_(Crhr2),_mRNA	2,41E-11	100	33
		Mm_RIKEN_cDNA_C030016D13_gene_(C030016D13Rik),_non-coding_RNA	0,34	91,67	24
		Mm_family_with_sequence_similarity_73,_member_B_(Fam73b),_transcript_variant_1,_	0,34	100	16
Cxcr7	For	Mm_chemokine_(C-X-C_motif)_receptor_7_(Cxcr7),_mRNA	8,51E-22	100	51
		Mm_glucocorticoid_receptor_DNA_binding_factor_1_(Grif1),_mRNA	0,17	100	17
		Mm_pleckstrin_homology_domain_containing,_family_N_member_1_(Plekhn1),_mRNA	0,17	100	17
	Rev	Mm_chemokine_(C-X-C_motif)_receptor_7_(Cxcr7),_mRNA	4,80E-20	100	48
		Mm_G-protein_coupled_receptor_12_(Gpr12),_transcript_variant_1,_mRNA	0,6	100	16
		Mm_armadillo_repeat_containing_3_(Armc3),_mRNA	0,6	100	16
Ednra	For	Mm_endothelin_receptor_type_A_(Ednra),_mRNA	2,83E-06	100	24
		Mm_family_with_sequence_similarity_13,_member_A_(Fam13a),_mRNA	0,043	100	17
		Mm_protein_phosphatase_1J_(Ppm1j),_mRNA	0,67	94,74	19
	Rev	Mm_endothelin_receptor_type_A_(Ednra),_mRNA	6,45E-12	100	34
		Mm_transmembrane_protein_132B_(Tmem132b),_mRNA	0,36	100	16
		Mm_zinc_finger_protein_382_(Zfp382),_mRNA	5,6	94,44	18
F2r	For	Mm_coagulation_factor_II_(thrombin)_receptor_(F2r),_mRNA	1,17E-04	100	21
		Mm_armadillo_repeat_containing_8_(Armc8),_transcript_variant_2,_mRNA	0,11	100	16
		Mm_armadillo_repeat_containing_8_(Armc8),_transcript_variant_1,_mRNA	0,11	100	16
	Rev	Mm_coagulation_factor_II_(thrombin)_receptor_(F2r),_mRNA	9,95E-06	100	23
		Mm_DNA_segment,_Chr_1,_ERATO_Doi_622,_expressed_(D1Erd622e),_mRNA	0,038	100	17
		Mm_small_nuclear_ribonucleoprotein_200_(U5)_(_Snrnp200),_mRNA	0,15	100	16
F2rl1	For	Mm_coagulation_factor_II_(thrombin)_receptor-like_1_(F2rl1),_mRNA	1,25E-20	100	49
		Mm_akirin_1_(Akirin1),_mRNA	0,62	100	16
		Mm_predicted_gene_5622_(Gm5622),_mRNA	2,4	94,74	19
	Rev	Mm_coagulation_factor_II_(thrombin)_receptor-like_1_(F2rl1),_mRNA	3,27E-21	100	50
		Mm_zinc_finger_protein_740_(Zfp740),_mRNA	0,64	100	16
		Mm_v-erb-erythroblastic_leukemia_viral_oncogene_homolog_2,_neuro/glioblastoma_	0,64	100	16
Gabbr1	For	Mm_gamma-aminobutyric_acid_(GABA)_B_receptor,_1_(Gabbr1),_mRNA	3,35E-10	100	31
		Mm_calcium_sensing_receptor_(Casr),_mRNA	4,7	100	14
		Mm_B_lymphoid_kinase_(Blk),_mRNA	4,7	100	14
	Rev	Mm_gamma-aminobutyric_acid_(GABA)_B_receptor,_1_(Gabbr1),_mRNA	2,45E-08	96,77	31
		Mm_beta_galactoside_alpha_2,6_sialyltransferase_1_(St6gal1),_transcript_variant_2,_m	1,4	100	15
		Mm_beta_galactoside_alpha_2,6_sialyltransferase_1_(St6gal1),_transcript_variant_1,_m	1,4	100	15
Gpr107	For	Mm_G_protein-coupled_receptor_107_(Gpr107),_mRNA	1,63E-12	100	35
		Mm_DiGeorge_syndrome_critical_region_gene_8_(Dgcr8),_mRNA	0,36	100	16
		Mm_GNAS_(guanine_nucleotide_binding_protein,_alpha_stimulating)_complex_locus_(1,4	100	15
	Rev	Mm_G_protein-coupled_receptor_107_(Gpr107),_mRNA	1,94E-04	100	21
		PREDICTED:_Mm_predicted_gene_10775_(Gm10775),_miscRNA	2,9	94,44	18
		Mm_SMC_hinge_domain_containing_1_(Smchd1),_mRNA	2,9	100	14
Gpr108	For	Mm_G_protein-coupled_receptor_108_(Gpr108),_mRNA	4,58E-09	100	29
		Mm_a_disintegrin-like_and_metallopeptidase_(reprolysin_type)_with_thrombospondin	0,017	100	18
		Mm_adenosine_monophosphate_deaminase_3_(Ampd3),_mRNA	0,066	100	17
	Rev	Mm_G_protein-coupled_receptor_108_(Gpr108),_mRNA	0,056	100	16
		Mm_microRNA_1198_(Mir1198),_microRNA	3,5	100	13
		Mm_desmocollin_1_(Dsc1),_mRNA	3,5	100	13

Gpr124	For	Mm_G_protein-coupled_receptor_124_(Gpr124),_mRNA	2,83E-06	100	24
		Mm_twisted_gastrulation_homolog_1_(Drosophila)_(Twsg1),_mRNA	0,17	100	16
		Mm_RIKEN_cDNA_2310065F04_gene_(2310065F04Rik),_non-coding_RNA	0,17	100	16
Rev		Mm_G_protein-coupled_receptor_124_(Gpr124),_mRNA	3,02E-07	100	26
		Mm_RNA_binding_protein,_fox-1_homolog_(C_elegans)_2_(Rbfox2),_transcript_variant	1,1	100	15
		Mm_RNA_binding_protein,_fox-1_homolog_(C_elegans)_2_(Rbfox2),_transcript_variant	1,1	100	15
Gpr125	For	Mm_G_protein-coupled_receptor_125_(Gpr125),_mRNA	1,17E-04	100	21
		Mm_chloride_intracellular_channel_4_(mitochondrial)_(Clic4),_nuclear_gene_encoding_	0,44	100	15
		Mm_RIKEN_cDNA_E130309D14_gene_(E130309D14Rik),_mRNA	1,8	100	14
Rev		Mm_G_protein-coupled_receptor_125_(Gpr125),_mRNA	3,35E-10	100	31
		PREDICTED:_Mm_predicted_gene_16258_(Gm16258),_miscRNA	0,3	100	16
		PREDICTED:_Mm_predicted_gene_16258_(Gm16258),_miscRNA	0,3	100	16
Gpr133	For	Mm_G_protein-coupled_receptor_133_(Gpr133),_mRNA	9,1	100	12
		Mm_Kell_blood_group_precursor_(McLeod_phenotype)_homolog_(Xk),_mRNA	9,1	100	12
		PREDICTED:_Mm_predicted_gene_16244_(Gm16244),_miscRNA	9,1	100	12
Rev		Mm_G_protein-coupled_receptor_133_(Gpr133),_mRNA	4,58E-09	100	29
		Mm_RIKEN_cDNA_0610007P08_gene_(0610007P08Rik),_transcript_variant_2,_mRNA	1	100	15
		Mm_cDNA_sequence_BC017643_(BC017643),_transcript_variant_5,_mRNA	4,1	100	14
Gpr137	For	Mm_G_protein-coupled_receptor_137_(Gpr137),_transcript_variant_1,_mRNA	1,68E-08	100	28
		Mm_G_protein-coupled_receptor_137_(Gpr137),_transcript_variant_2,_mRNA	1,68E-08	100	28
		Mm_G_protein-coupled_receptor_137_(Gpr137),_transcript_variant_4,_mRNA	1,68E-08	100	28
Rev		Mm_G_protein-coupled_receptor_137_(Gpr137),_transcript_variant_1,_mRNA	4,58E-09	100	29
		Mm_G_protein-coupled_receptor_137_(Gpr137),_transcript_variant_2,_mRNA	4,58E-09	100	29
		Mm_G_protein-coupled_receptor_137_(Gpr137),_transcript_variant_4,_mRNA	4,58E-09	100	29
Gpr137b	For	Mm_G_protein-coupled_receptor_137B_(Gpr137b),_mRNA	5,64E-16	100	41
		Mm_G_protein-coupled_receptor_137B,_pseudogene_(Gpr137b-ps),_non-coding_RNA	5,64E-16	100	41
		PREDICTED:_Mm_uncharacterized_LOC100504746,_transcript_variant_1_(LOC10050474	7,3	100	14
Rev		Mm_G_protein-coupled_receptor_137B_(Gpr137b),_mRNA	6,88E-05	96,67	30
		Mm_G_protein-coupled_receptor_137B,_pseudogene_(Gpr137b-ps),_non-coding_RNA	0,017	93,33	30
		Mm_ATPase,_class_V,_type_10B_(Atp10b),_mRNA	1	100	15
Gpr153	For	Mm_G_protein-coupled_receptor_153_(Gpr153),_mRNA	1,68E-08	100	28
		Mm_PRP38_pre-mRNA_processing_factor_38_(yeast)_domain_containing_B_(Prpf38b),	0,96	100	15
		Mm_retinoblastoma_binding_protein_8_(Rbbp8),_transcript_variant_3,_non-coding_R	3,8	100	14
Rev		Mm_G_protein-coupled_receptor_153_(Gpr153),_mRNA	2,41E-11	100	33
		Mm_predicted_gene_11696_(Gm11696),_transcript_variant_1,_non-coding_RNA	0,34	100	16
		Mm_exonuclease_3'-5'_domain_containing_2_(Exd2),_mRNA	1,3	100	15
Gpr21	For	Mm_G_protein-coupled_receptor_21_(Gpr21),_mRNA	2,68E-18	100	45
		Mm_RIKEN_cDNA_9430014N10_gene_(9430014N10Rik),_non-coding_RNA	0,009	100	19
		Mm_patched_domain_containing_3_(Ptchd3),_mRNA	0,035	100	18
Rev		Mm_G_protein-coupled_receptor_21_(Gpr21),_mRNA	3,27E-21	100	50
		Mm_zinc_finger_protein_317_(Zfp317),_mRNA	0,64	100	16
		Mm_glutamine_fructose-6-phosphate_transaminase_1_(Gfpt1),_mRNA	0,64	100	16
Gpr4	For	Mm_G_protein-coupled_receptor_4_(Gpr4),_mRNA	3,21E-15	94,23	52
		Mm_integrin_alpha_X_(Itgax),_mRNA	0,17	100	17
		PREDICTED:_Mms_predicted_gene_10723_(Gm10723),_miscRNA	0,67	100	16
Rev		Mm_G_protein-coupled_receptor_4_(Gpr4),_mRNA	1,92E-13	93,88	49
		Mm_heat_shock_protein_5_(Hspa5),_transcript_variant_2,_mRNA	0,17	100	17
		PREDICTED:_Mm_predicted_gene_10723_(Gm10723),_miscRNA	0,65	100	16
Gpr64	For	Mm_G_protein-coupled_receptor_64_(Gpr64),_transcript_variant_4,_mRNA	7,97E-07	100	25
		Mm_G_protein-coupled_receptor_64_(Gpr64),_transcript_variant_3,_mRNA	7,97E-07	100	25
		Mm_G_protein-coupled_receptor_64_(Gpr64),_transcript_variant_2,_mRNA	7,97E-07	100	25
Rev		Mm_G_protein-coupled_receptor_64_(Gpr64),_transcript_variant_4,_mRNA	1,68E-08	100	28
		Mm_G_protein-coupled_receptor_64_(Gpr64),_transcript_variant_3,_mRNA	1,68E-08	100	28
		Mm_G_protein-coupled_receptor_64_(Gpr64),_transcript_variant_2,_mRNA	1,68E-08	100	28
Gprc5b	For	Mm_G_protein-coupled_receptor,_family_C,_group_5,_member_B_(Gprc5b),_transcript	2,68E-18	100	45
		Mm_G_protein-coupled_receptor,_family_C,_group_5,_member_B_(Gprc5b),_transcript	2,68E-18	100	45
		Mm_nuclear_apoptosis_inducing_factor_1_(Naif1),_mRNA	0,54	100	16
Rev		Mm_G_protein-coupled_receptor,_family_C,_group_5,_member_B_(Gprc5b),_transcript	4,80E-20	100	48
		Mm_G_protein-coupled_receptor,_family_C,_group_5,_member_B_(Gprc5b),_transcript	4,80E-20	100	48
		Mm_SNF_related_kinase_(Snrk),_transcript_variant_2,_mRNA	0,038	100	18
Htr2a	For	Mm_5-hydroxytryptamine_(serotonin)_receptor_2A_(Htr2a),_mRNA	9,01E-11	100	32
		Mm_Sec1_family_domain_containing_2_(Sctd2),_transcript_variant_b,_mRNA	0,02	100	18

		Mm_Sec1_family_domain_containing_2_(Scfd2),_transcript_variant_a,_mRNA	0,02	100	18
Rev		Mm_5-hydroxytryptamine_(serotonin)_receptor_2A_(Htr2a),_mRNA	1,63E-12	100	35
		PREDICTED:_Mm_predicted_gene_9930_(Gm9930),_miscRNA	1,4	100	15
		Mm_cDNA_sequence_BC027231_(BC027231),_mRNA	1,4	100	15
Lgr4	For	Mm_leucine-rich_repeat-containing_G_protein-coupled_receptor_4_(Lgr4),_mRNA	9,01E-11	100	32
		Mm_small_G_protein_signaling_modulator_2_(Sgsm2),_mRNA	0,32	100	16
		Mm_solute_carrier_family_16_(monocarboxylic_acid_transporters),_member_11_(Slc16	1,3	100	15
Rev		Mm_leucine-rich_repeat-containing_G_protein-coupled_receptor_4_(Lgr4),_mRNA	3,35E-10	100	31
		Mm_acyl-CoA_thioesterase_5_(Acot5),_mRNA	0,3	100	16
		Mm_acyl-CoA_thioesterase_3_(Acot3),_mRNA	0,3	100	16
Lgr6	For	Mm_leucine-rich_repeat-containing_G_protein-coupled_receptor_6_(Lgr6),_mRNA	9,01E-11	100	32
		Mm_phosphoinositide-3-kinase,_regulatory_subunit_5,_p101_(Pik3r5),_mRNA	1,3	100	15
		Mm_leucine-rich_repeat_containing_38_(Lrrc38),_mRNA	1,3	100	15
Rev		Mm_leucine-rich_repeat-containing_G_protein-coupled_receptor_6_(Lgr6),_mRNA	1,63E-12	100	35
		Mm_SCO_cytochrome_oxidase_deficient_homolog_1_(yeast)_(_Sco1),_nuclear_gene_enc	0,36	100	16
		PREDICTED:_Mms_predicted_gene_6934_(Gm6934),_mRNA	1,4	100	15
Lpar1	For	Mm_lysofosfpatidic_acid_receptor_1_(Lpar1),_transcript_variant_2,_mRNA	1,84E-19	100	47
		Mm_lysofosfpatidic_acid_receptor_1_(Lpar1),_transcript_variant_1,_mRNA	1,84E-19	100	47
		Mm_dopamine_receptor_D3_(Drd3),_mRNA	0,002	90,63	32
Rev		Mm_lysofosfpatidic_acid_receptor_1_(Lpar1),_transcript_variant_2,_mRNA	3,90E-17	100	43
		Mm_lysofosfpatidic_acid_receptor_1_(Lpar1),_transcript_variant_1,_mRNA	3,90E-17	100	43
		Mm_pyruvate_kinase_liver_and_red_blood_cell_(Pklr),_nuclear_gene_encoding_mitoch	0,13	100	17
Lpar4	For	Mm_lysofosfpatidic_acid_receptor_4_(Lpar4),_mRNA	2,68E-18	100	45
		PREDICTED:_Mm_predicted_gene_13660_(Gm13660),_miscRNA	2,1	100	15
		Mm_interferon_(alpha_and_beta)_receptor_2_(Ifnar2),_transcript_variant_1,_mRNA	2,1	100	15
Rev		Mm_lysofosfpatidic_acid_receptor_4_(Lpar4),_mRNA	1,25E-20	100	49
		Mm_additional_sex_combs_like_1_(Drosophila)_(_Asxl1),_mRNA	0,62	95	20
		Mm_ATPase_type_13A3_(Atp13a3),_transcript_variant_1,_mRNA	2,4	100	15
Lphn1	For	Mm_latrophilin_1_(Lphn1),_mRNA	2,32E-15	100	40
		PREDICTED:_Mm_predicted_gene_10644_(Gm10644),_miscRNA	2,32E-15	100	40
		PREDICTED:_Mm_predicted_gene_10644_(Gm10644),_miscRNA	2,32E-15	100	40
Rev		Mm_latrophilin_1_(Lphn1),_mRNA	9,69E-08	97,14	35
		PREDICTED:_Mm_predicted_gene_10644_(Gm10644),_miscRNA	9,69E-08	97,14	35
		PREDICTED:_Mm_predicted_gene_10644_(Gm10644),_miscRNA	9,69E-08	97,14	35
Lphn2	For	Mm_latrophilin_2_(Lphn2),_mRNA	2,68E-11	100	33
		Mm_tetratricopeptide_repeat_domain_16_(Ttc16),_mRNA	0,095	100	17
		Mm_sel-1_suppressor_of_lin-12-like_3_(C._elegans)_(_Sel1l3),_mRNA	0,37	100	16
Rev		Mm_latrophilin_2_(Lphn2),_mRNA	1,24E-09	100	30
		Mm_caspase_8_associated_protein_2_(Casp8ap2),_transcript_variant_2,_mRNA	1,1	94,74	19
		Mm_caspase_8_associated_protein_2_(Casp8ap2),_transcript_variant_1,_mRNA	1,1	94,74	19
Mrgprf	For	Mm_MAS-related_GPR,_member_F_(Mrgprf),_mRNA	2,22E-07	100	26
		Mm_copine_II_(Cpne2),_mRNA	0,052	100	17
		Mm_FERM,_RhoGEF_(Arhgef)_and_pleckstrin_domain_protein_1_(chondrocyte-derived	0,81	100	15
Rev		Mm_MAS-related_GPR,_member_F_(Mrgprf),_mRNA	2,45E-08	100	28
		Mm_lectin,_mannose-binding_2_(Lman2),_mRNA	0,36	100	16
		Mm_RIKEN_cDNA_9430015G10_gene_(9430015G10Rik),_transcript_variant_2,_mRNA	0,36	100	16
Npy1r	For	Mm_neuropeptide_Y_receptor_Y1_(Npy1r),_mRNA	3,35E-10	100	31
		Mm_collagen_type_XVIII,_alpha_1_(Col18a1),_transcript_variant_1,_mRNA	0,3	95	20
		Mm_collagen_type_XVIII,_alpha_1_(Col18a1),_transcript_variant_2,_mRNA	0,3	95	20
Rev		Mm_neuropeptide_Y_receptor_Y1_(Npy1r),_mRNA	3,06E-14	100	38
		Mm_PHD_finger_protein_23_(Phf23),_mRNA	0,41	100	16
		PREDICTED:_Mm_predicted_gene_5524_(Gm5524),_miscRNA	1,6	100	15
P2ry2	For	Mm_purinergic_receptor_P2Y,_G-protein_coupled_2_(P2ry2),_mRNA	2,83E-06	100	24
		Mm_cadherin-like_24_(Cdh24),_mRNA	0,67	100	15
		Mm_dehydrogenase/reductase_(SDR_family)_member_11_(Dhrs11),_mRNA	0,67	100	15
Rev		Mm_purinergic_receptor_P2Y,_G-protein_coupled_2_(P2ry2),_mRNA	1,15E-13	100	37
		PREDICTED:_Mm_myosin-6-like_(LOC100862557),_mRNA	1,6	100	15
		Mm_potassium_channel_subfamily_K,_member_7_(Kcnk7),_mRNA	1,6	100	15
P2ry6	For	Mm_pyrimidinergic_receptor_P2Y,_G-protein_coupled,_6_(P2ry6),_mRNA	2,22E-07	100	26
		Mm_nucleoporin_210-like_(Nup210l),_mRNA	0,21	100	16
		Mm_ribosomal_protein_L18A_(Rpl18a),_mRNA	0,81	100	15
Rev		Mm_pyrimidinergic_receptor_P2Y,_G-protein_coupled,_6_(P2ry6),_mRNA	2,41E-11	100	33

		PREDICTED: Mm_ring_finger_protein_213_(Rnf213),_mRNA	1,3	100	15
		Mm_SID1_transmembrane_family_member_1_(Sid1),_transcript_variant_1,_mRNA	1,3	100	15
Ptger3	For	Mm_prostaglandin_E_receptor_3_(subtype_EP3)(Ptger3),_mRNA	1,25E-20	100	49
		Mm_solute_carrier_family_9_(sodium/hydrogen_exchanger),_member_9_(Slc9a9),_mRNA	0,04	100	18
		Mm_ATP-binding_cassette_sub-family_A_(ABC1),_member_15_(Abca15),_mRNA	0,62	95	20
	Rev	Mm_prostaglandin_E_receptor_3_(subtype_EP3)(Ptger3),_mRNA	1,84E-19	100	47
		Mm_component_of_oligomeric_golgi_complex_2_(Cog2),_mRNA	2,3	100	15
		Mm_solute_carrier_family_26_(sulfate_transporter),_member_1_(Slc26a1),_mRNA	2,3	100	15
Ptgir	For	Mm_prostaglandin_I_receptor_(IP)(Ptgir),_mRNA	7,97E-07	100	25
		Mm_SET_domain_containing_(lysine_methyltransferase)_7_(Setd7),_mRNA	0,19	95	20
		Mm_RIKEN_cDNA_4921511C10_gene_(4921511C10Rik),_non-coding_RNA	0,74	100	15
	Rev	Mm_prostaglandin_I_receptor_(IP)(Ptgir),_mRNA	8,10E-15	100	39
		Mm_solute_carrier_family_30_(zinc_transporter),_member_5_(Slc30a5),_mRNA	0,11	95,24	21
		Mm_nucleoporin_85_(Nup85),_mRNA	0,43	95	20
Pth1r	For	Mm_parathyroid_hormone_1_receptor_(Pth1r),_transcript_variant_3,_mRNA	3,06E-14	100	38
		Mm_parathyroid_hormone_1_receptor_(Pth1r),_transcript_variant_2,_mRNA	3,06E-14	100	38
		Mm_parathyroid_hormone_1_receptor_(Pth1r),_transcript_variant_1,_mRNA	3,06E-14	100	38
	Rev	Mm_parathyroid_hormone_1_receptor_(Pth1r),_transcript_variant_3,_mRNA	6,21E-09	100	29
		Mm_parathyroid_hormone_1_receptor_(Pth1r),_transcript_variant_2,_mRNA	6,21E-09	100	29
		Mm_parathyroid_hormone_1_receptor_(Pth1r),_transcript_variant_1,_mRNA	6,21E-09	100	29
Tbxa2r	For	Mm_thromboxane_A2_receptor_(Tbxa2r),_mRNA	2,68E-18	100	45
		Mm_prostaglandin_E_receptor_1_(subtype_EP1)(Ptger1),_mRNA	9,13E-06	100	24
		PREDICTED: Mm_RIKEN_cDNA_1700040D17_gene_(1700040D17Rik),_miscRNA	0,54	100	16
	Rev	Mm_thromboxane_A2_receptor_(Tbxa2r),_mRNA	2,41E-11	100	33
		Mm_glutamic-oxaloacetic_transaminase_1-like_1_(Got1l1),_mRNA	1,3	100	15
		Mm_a_disintegrin_and_metallopeptidase_domain_1a_(Adam1a),_mRNA	1,3	100	15

Table 2: List of blast hits for sequences obtained by sequencing of single-cell amplicons using forward (For) or reverse (Rev) primers (only the three highest ranking hits are shown per analysis). The E-value describes the number of hits expected to see by chance when searching a database of a comparable size. An E-value of 1 indicates that in a database of the current size one might expect to see 1 match with a similar score simply by chance. Mm, mus musculus.

To summarize these findings, we added the following paragraph on page 7, line 17-20 of the revised manuscript:

“We also sequenced single-cell RT-PCR amplicons to exclude off-target amplification or amplification of highly homologous GPCRs and found that the amplified sequences were in all cases specific for the targeted receptor (data not shown).”

We furthermore mention the amplicon sequencing results in the discussion on page 14, line 15.

2. The authors found that different types of SMCs or ECs have distinct GPCR expression patterns and can be grouped into distinct clusters by clustering analysis. The authors should give a list to clearly show how many experiments and how many cells in each experiment have been analyzed for each SMC and EC cell types. In Figure 3G and 4C, for a certain cluster, have the cells been analyzed by more than one independent experiments to exclude the batch effect?

Following the reviewer’s suggestion we assembled a table showing for each cell type the number of cells, mice, and independent experiments (Table 3).

Cell type	No. of cells	No. of mice	No. of exps
SMao	60	8	3
SMaoAged	32	6	2
SMaoApo0w	34	6	2
SMaoApo16w	50 (prev. vers.: 57)	6	2
SMsk	57 (prev. vers.: 66)	7	3
SMmes	29	8	5
SMub	25	8	4
ECsk	40	6	5
EClu	48	8	3
ECluLPS	25	4	2
ECbr	52	7	2
ECbrLPS	22	4	2
ECao	12	4	2
ECaoApo16w	16	6	2

Table 3: Overview of numbers of cells, mice, and independent experiments analyzed for the different cell types.

As shown in Table 3, data underlying cluster analyses in Figures 3G and 4C were generated in 2-5 independent experiments and 6-8 mice per group.

We added the total numbers of cells and mice to the respective Figure legends of the revised manuscript, in addition we attached Table 3 of this letter as Supplemental Figure 4 to the revised manuscript and refer to it on page 9, lines 1-2.

The numbers of SMsk and SMaoApo16w were unfortunately incorrect in the previous version of the manuscript, these mistakes have been corrected in Figure legends 1, 3 and 7 of the revised manuscript (for SMsk 57 instead of 66, for SMaoApo16w from 50 instead of 57).

3. The authors also found that GPCR expression was highly heterogeneous within each cell type. However, in most conditions, it seems that the authors have not identified functional cell subpopulations. I am not convinced that the intra-cell type heterogeneous GPCR expression has important biological functions in general. The authors should be careful to claim this. It can be helpful to verify the heterogeneous GPCR expression at the protein level if suitable antibodies are available. If you can detect the heterogeneously expressed GPCR gene at the protein level, it is more convincing that the gene could be functional.

We followed the reviewer's suggestion and investigated GPCR heterogeneity in aortic smooth muscle cells on the protein level by flow cytometry. To do so, aortae of wildtype mice were digested and permeabilized as described in the methods section, followed by incubation of single cell suspensions with antibodies directed against α SMA (PE or FITC) and APC-labelled antibodies directed against receptors Cmk1r1, Ccr12, Celsr2, and Cxcr7. For each GPCR-specific antibody the corresponding isotype control was used (Fig. 2A). We found that also on the protein level heterogeneity of GPCR was present within the SMao population, and that the percentages roughly matched the values obtained by single-cell RT-PCR (Fig. 2B).

Figure 2: Flow cytometric analysis of GPCR expression. **A**, Example of the gating strategy. **B**, GPCR expression frequency in individual SMao as judged by single-cell flow cytometric analysis (antibody/FACS) or RT-PCR (sc RT-PCR).

We included these data in Suppl. Figures 2B and C and describe them on page 7, lines 20-23 of the revised manuscript as well as on page 22 of the revised Supplemental Material. We also mention these results in the discussion on page 14, lines 15 and 23.

4. The authors have identified a small subpopulation of dedifferentiating SMC from healthy SMao (Figure 8). They have also performed the single cell RNA-seq method for SMao. Could this small subpopulation also be identified from the single cell RNA-seq data and were the results between the single cell RT-PCR and single cell RNA-seq consistent?

We followed the reviewers' suggestion and investigated whether the small subpopulation of spontaneously dedifferentiating SMao was also found in the mRNAseq data. To do so, we applied the algorithm described by Grün et al¹ to the SMao mRNAseq data set, which led to the identification a subgroup of 14% of cells with distinct expression pattern (Fig.3A). This subgroup showed features indicative of a dedifferentiated state, such as expression of receptors or transcription factors that have been implicated in the regulation of smooth muscle (de-)differentiation, such as *Klf4*^{2, 3, 4, 5}, *Notch1*^{6, 7, 8}, or *FGF* receptors^{9, 10}. Furthermore, this population of cells showed a reduced expression of *Tagln2*, a homologue of the smooth muscle differentiation marker transgelin, also known as *Sm22a*¹¹ (Fig. 3B). Other markers typically changed in dedifferentiating smooth muscle cells, such as *Icam1* or *Vcam1*, were not enriched in these cells. This was probably due to the reduced detection frequency in mRNAseq compared to RTPCR: *Vcam1* was only detected in 2% of cells (RT-PCR: 10%) and *Icam1* not at all (RT-PCR: 8%). However, assuming that the above-mentioned subgroup represents a dedifferentiating subpopulation, we next investigated how their GPCR pattern differed. Cluster analysis identified only six GPCRs mRNAs that were significantly increased in the subpopulation (Fig. 3C): those encoding the orphan receptors *Gpr20* and *Mrgprh*, the olfactory receptors *Olfcr558* and *Olfcr78*, the corticotrophin releasing hormone receptors subtype 2 (*Crhr2*), and the endothelin receptor subtype A (*Ednra*). Three of these GPCRs were not included in the RT-PCR screen (*Gpr20*, *Olfcr558*, *Mrgprh*); of the remaining three, *Ednra* and *Olfcr78* were also in RT-PCR analysis enriched in dedifferentiating SMC (either spontaneous or atherosclerotic). However, the majority of GPCRs that were in RT-PCR found to be associated with a dedifferentiated state were not enriched in the putative SMaoDiff in mRNAseq. Again, this is most likely due differences in detection frequency, since all GPCRs identified in RT-PCR were, except for *Lgr6*, less efficiently detected in single-cell mRNAseq (Fig. 3D). We therefore conclude that the difference in detection frequency of low

abundance transcripts such as GPCRs hinders the direct comparison of data from single-cell RT-PCR and mRNAseq.

Figure 3: Functional subgroups within the SMao mRNAseq data set. **A**, T-SNE plot of k-means clustering data: cluster assignment is indicated by colored numbers (each dot one cell; distance between dots indicates degree of similarity). **B,C**, Fold change of gene expression in cluster 1 cells compared to the rest of SMao: genes indicative of differentiation state (B) and GPCRs (C). **D**, Comparison of the expression frequency in mRNAseq versus RT-PCR for those GPCRs that are according to single-cell RT-PCR enriched in dedifferentiating SMao.

Other changes

- We noted that the order of transcription factors in Figure 7F was incorrect, this has been remedied.
- The last sentence of paragraph 1 of the discussion (“However, it cannot be excluded that also with the RT-PCR method very low abundance transcripts escape detection”) was deleted since it was largely redundant with the sentence in lines 9-11 of the same paragraph.
- The number of function-defining genes included in the array has been corrected from 12 to 13 genes, since in the original version *Lyve1* (for lymphatic EC) was left out. Also the number of function-defining genes has been corrected to 36.
- Since the term “expression” was used twice in the title, we renamed the manuscript as follows: “Single-cell profiling reveals heterogeneity and functional patterning of GPCR expression in the vascular system” (instead of Single-cell expression profiling reveals heterogeneity and functional patterning of GPCR expression in the vascular system).

References

1. Grun D, *et al.* Single-cell messenger RNA sequencing reveals rare intestinal cell types. *Nature* **525**, 251-255 (2015).
2. Shankman LS, *et al.* KLF4-dependent phenotypic modulation of smooth muscle cells has a key role in atherosclerotic plaque pathogenesis. *Nat Med* **21**, 628-637 (2015).
3. Yoshida T, Kaestner KH, Owens GK. Conditional deletion of Kruppel-like factor 4 delays downregulation of smooth muscle cell differentiation markers but accelerates neointimal formation following vascular injury. *Circ Res* **102**, 1548-1557 (2008).
4. Yoshida T, Gan Q, Owens GK. Kruppel-like factor 4, Elk-1, and histone deacetylases cooperatively suppress smooth muscle cell differentiation markers in response to oxidized phospholipids. *Am J Physiol Cell Physiol* **295**, C1175-1182 (2008).
5. Liu Y, Sinha S, McDonald OG, Shang Y, Hoofnagle MH, Owens GK. Kruppel-like factor 4 abrogates myocardin-induced activation of smooth muscle gene expression. *J Biol Chem* **280**, 9719-9727 (2005).
6. Li Y, *et al.* Smooth muscle Notch1 mediates neointimal formation after vascular injury. *Circulation* **119**, 2686-2692 (2009).
7. Proweller A, Pear WS, Parmacek MS. Notch signaling represses myocardin-induced smooth muscle cell differentiation. *J Biol Chem* **280**, 8994-9004 (2005).
8. Boucher J, Gridley T, Liaw L. Molecular pathways of notch signaling in vascular smooth muscle cells. *Front Physiol* **3**, 81 (2012).
9. Chen PY, Qin L, Li G, Tellides G, Simons M. Fibroblast growth factor (FGF) signaling regulates transforming growth factor beta (TGFbeta)-dependent smooth muscle cell phenotype modulation. *Sci Rep* **6**, 33407 (2016).
10. Chen PY, Qin L, Li G, Tellides G, Simons M. Smooth muscle FGF/TGFbeta cross talk regulates atherosclerosis progression. *EMBO Mol Med* **8**, 712-728 (2016).
11. Robin YM, *et al.* Transgelin is a novel marker of smooth muscle differentiation that improves diagnostic accuracy of leiomyosarcomas: a comparative immunohistochemical reappraisal of myogenic markers in 900 soft tissue tumors. *Mod Pathol* **26**, 502-510 (2013).

	Mus_musculus_shroom_family_member_4_(Shroom4),_mRNA	0.54	100	16	17	32	CATGGTGGTGGCCAGC
	Mus_musculus_chemokine_(C-C_motif)_receptor_9_(Ccr9),_transcript_variant_2,_mRNA	0.54	100	16	9	24	GTGGGCATCATGGTGG
	Mus_musculus_chemokine_(C-C_motif)_receptor_9_(Ccr9),_transcript_variant_1,_mRNA	0.54	100	16	9	24	GTGGGCATCATGGTGG
	PREDICTED:_Mus_musculus_RIKEN_cDNA_1700040D17_gene_(1700040D17Rik),_miscRNA	0.54	100	16	20	35	GGTGGTGGCCACGGTG
	PREDICTED:_Mus_musculus_predicted_pseudogene_5958_(Gm5958),_mRNA	2.1	100	15	16	30	TCATGGTGGTGGCCCA
	PREDICTED:_Mus_musculus_cell_division_cycle_20_homolog_B_(S_cerevisiae)_(Cdc20b),_mRNA	2.1	94.74	19	14	32	CATCATGGTGGTGGCCACG
	PREDICTED:_Mus_musculus_predicted_pseudogene_5958_(Gm5958),_mRNA	2.1	100	15	16	30	TCATGGTGGTGGCCCA
	PREDICTED:_Mus_musculus_cell_division_cycle_20_homolog_B_(S_cerevisiae)_(Cdc20b),_mRNA	2.1	94.74	19	14	32	CATCATGGTGGTGGCCACG
	Mus_musculus_zinc_finger_protein_748_(Zfp748),_transcript_variant_2,_mRNA	2.1	100	15	17	31	CATGGTGGTGGCCAC
	Mus_musculus_zinc_finger_protein_748_(Zfp748),_transcript_variant_1,_mRNA	2.1	100	15	17	31	CATGGTGGTGGCCAC
	PREDICTED:_Mus_musculus_predicted_gene_9837_(Gm9837),_miscRNA	2.1	100	15	23	37	GGTGGCCACGGTGTG
	Mus_musculus_gap_junction_protein_delta_3_(Gjd3),_mRNA	2.1	91.3	23	13	35	GCATCATGGTGGTGGCCACGGTG
	Mus_musculus_heparan_sulfate_(glucosamine)_3-O-sulfotransferase_3B1_(Hs3st3b1),_mRNA	2.1	100	15	6	20	CTCGTGGGCATCATG
	PREDICTED:_Mus_musculus_predicted_gene_9837_(Gm9837),_miscRNA	2.1	100	15	23	37	GGTGGCCACGGTGTG
	Mus_musculus_SRY-box_containing_gene_6_(Sox6),_transcript_variant_1,_mRNA	2.1	100	15	17	31	CATGGTGGTGGCCAC
	Mus_musculus_multiple_EGF-like_domains_9_(Megf8),_mRNA	2.1	100	15	24	38	GTGGCCACGGTGTGT
	Mus_musculus_contactin_associated_protein-like_2_(Cntnap2),_transcript_variant_1,_mRNA	2.1	100	15	25	39	TGGCCACGGTGTGT
Tbxa2r_F	Mus_musculus_thromboxane_A2_receptor_(Tbxa2r),_mRNA	2.41E-11	100	33	1	33	TTCATGCAGACTTTGTGCAGACACCCT
	Mus_musculus_glutamic-oxaloacetic_transaminase_1-like_1_(Got1l1),_mRNA	1.3	100	15	5	19	TCATGCAGACTTTGT
	Mus_musculus_a_disintegrin_and_metalloproteinase_domain_1a_(Adam1a),_mRNA	1.3	100	15	6	20	CATGCAGACTTTGT
	Mus_musculus_nucleolar_and_spindle_associated_protein_1_(Nusap1),_transcript_variant_2,_mRNA	1.3	100	15	10	24	CAGACTTTGTGCAG
	Mus_musculus_nucleolar_and_spindle_associated_protein_1_(Nusap1),_transcript_variant_1,_mRNA	1.3	100	15	10	24	CAGACTTTGTGCAG
	Mus_musculus_regulatory_factor_X_3_(influences_HLA_class_II_expression)_(Rfx3),_transcript_variant_2,_mRNA	5.3	100	14	3	16	CATCATGCAGACTT
	Mus_musculus_regulatory_factor_X_3_(influences_HLA_class_II_expression)_(Rfx3),_transcript_variant_1,_mRNA	5.3	100	14	3	16	CATCATGCAGACTT
	Mus_musculus_fatty_acid_desaturase_3_(Fads3),_mRNA	5.3	100	14	17	30	TGTTGCAGACACCA
	Mus_musculus_Notch_gene_homolog_3_(Drosophila)_(Notch3),_mRNA	5.3	100	14	19	32	TTGCAGACACCAC
	Mus_musculus_transmembrane_protein_55b_(Tmem55b),_mRNA	5.3	100	14	18	31	GTTCAGACACCAC
	Mus_musculus_vomerinasal_1_receptor_200_(Vmn1r200),_mRNA	5.3	100	14	6	19	CATGCAGACTTTGT
	Mus_musculus_calmodulin_1_(Calm1),_mRNA	5.3	100	14	6	19	CATGCAGACTTTGT
	Mus_musculus_HEAT_repeat_containing_6_(Heatr6),_mRNA	5.3	100	14	15	28	TTTGTGCAGACAC
	Mus_musculus_RIKEN_cDNA_D930048N14_gene_(D930048N14Rik),_non-coding_RNA	5.3	100	14	1	14	TTCATCATGCAGAC
	Mus_musculus_netrin_4_(Ntn4),_mRNA	5.3	100	14	16	29	TTTGTGCAGACACC
	Mus_musculus_CD164_antigen_(Cd164),_mRNA	5.3	100	14	4	17	ATCATGCAGACTTT
	Mus_musculus_proprotein_convertase_subtilisin/kexin_type_7_(Pcsk7),_mRNA	5.3	100	14	2	15	TCATCATGCAGACTT
	Mus_musculus_phosphatidylinositol_glycan_anchor_biosynthesis_class_Y-like_(Pigy1),_mRNA	5.3	100	14	20	33	TGCAGACACCACCT
	Mus_musculus_piwi-like_homolog_4_(Drosophila)_(Pwi4),_mRNA	5.3	100	14	5	18	TCATGCAGACTTTG
	Mus_musculus_arginine/serine-rich_coiled-coil_2_(Rsrc2),_transcript_variant_3,_mRNA	5.3	100	14	16	29	TTTGTGCAGACACC

REVIEWERS' COMMENTS:

Reviewer #4 (Remarks to the Author):

My questions have mostly been answered. Just that the authors verified the protein-level expression of several GPCR receptors by flow cytometry. From my point of view, immunostaining is better that can directly see the in situ heterogeneity.